# Analysis of Selected Nutritional Parameters in Patients with HPV-Related and Non-HPV-Related Oropharyngeal Cancer before and after Radiotherapy Alone or Combined with Chemotherapy

**DOI:** 10.3390/cancers14092335

**Published:** 2022-05-09

**Authors:** Adam Brewczyński, Beata Jabłońska, Agnieszka Maria Mazurek, Jolanta Mrochem-Kwarciak, Sławomir Mrowiec, Mirosław Śnietura, Marek Kentnowski, Anna Kotylak, Zofia Kołosza, Krzysztof Składowski, Tomasz Rutkowski

**Affiliations:** 1I Radiation and Clinical Oncology Department, Maria Skłodowska-Curie National Research Institute of Oncology, Gliwice Branch, 44-102 Gliwice, Poland; adam.brewczynski@io.gliwice.pl (A.B.); marek.kentnowski@io.gliwice.pl (M.K.); anna.kotylak@io.gliwice.pl (A.K.); krzysztof.skladowski@io.gliwice.pl (K.S.); tomasz.rutkowski@io.gliwice.pl (T.R.); 2Department of Digestive Tract Surgery, Medical University of Silesia, 40-752 Katowice, Poland; mrowasm@poczta.onet.pl; 3Centre for Translational Research and Molecular Biology of Cancer, Maria Skłodowska-Curie National Research Institute of Oncology, Gliwice Branch, 44-102 Gliwice, Poland; agnieszka.mazurek@io.gliwice.pl; 4The Analytics and Clinical Biochemistry Department, Maria Skłodowska-Curie National Research Institute of Oncology, Gliwice Branch, 44-102 Gliwice, Poland; jolanta.mrochem-kwarciak@io.gliwice.pl; 5Tumor Pathology Department, Maria Skłodowska-Curie National Research Institute of Oncology, Gliwice Branch, 44-102 Gliwice, Poland; miroslaw.snietura@io.gliwice.pl; 6Department of Biostatistics and Bioinformatics, Maria Sklodowska-Curie National Research Institute of Oncology, Gliwice Branch, 44-102 Gliwice, Poland; zofia.kolosza@io.gliwice.pl

**Keywords:** oropharyngeal cancer, oropharyngeal carcinoma, human papilloma virus (HPV), nutritional status, malnutrition, radiotherapy, chemoradiotherapy

## Abstract

**Simple Summary:**

The aim of this study was to assess and compare the nutritional status (NS) of patients with HPV-related (HPV+) and non-HPV-related (HPV-) oropharyngeal cancer (OPC) before and after radiotherapy (RT) or chemoradiotherapy (CRT). The analysis included 127 patients with OPC who underwent radiotherapy (RT) alone, or in combination with chemotherapy (CRT). In both groups, a significant decrease in all analyzed nutritional parameters was noted after RT/CRT. Overall survival (OS) and disease-free survival (DFS) were significantly better in patients with a higher BMI in the HPV- group; DFS was significantly better in patients with total lymphocyte count (TLC) >1.28/mm^3^ in the HPV+ group. Higher NRS 2002 was an independent adverse prognostic factor for OS and DFS in HPV-, but not in the HPV+ group. Regardless of HPV status, patients with OPC can develop malnutrition during RT/CRT. Therefore, nutritional support during RT/CRT is required in patients with HPV- and HPV+ OPC.

**Abstract:**

Background: Radiotherapy plays an essential role in the treatment of oropharyngeal carcinoma (OPC). The aim of this study was to assess and compare the nutritional status (NS) of patients with HPV-related (HPV+) and non-HPV-related (HPV-) OPC before and after radiotherapy (RT) or chemoradiotherapy (CRT). Methods: The analysis included 127 patients with OPC who underwent radiotherapy (RT) alone, or in combination with chemotherapy (CRT), in the I Radiation and Clinical Oncology Department of Maria Skłodowska-Curie National Research Institute of Oncology, Gliwice Branch, Poland. Patients were divided according to HPV status. Confirmation of HPV etiology was obtained from FFPE (formalin-fixed, paraffin-embedded) tissue material and/or extracellular circulating HPV DNA. Basic anthropometric and biochemical parameters before and after RT/CRT were compared between the HPV- and HPV+ groups. The effect of NS on survival was also analyzed. Results: In both groups, a significant decrease in all analyzed nutritional parameters was noted after RT/CRT (*p* < 0.01). CRT caused significant weight loss and decreases in BMI, albumin, total lymphocyte count (TLC), and hemoglobin concentration, as well as an increase in the Nutritional Risk Score (NRS) 2002, in HPV- and HPV+ patients. A significant decrease in prealbumin levels after CRT was noted only in HPV+ patients. RT caused a significant decrease in hemoglobin concentration and TLC in HPV- patients. There were no significant differences regarding other nutritional parameters after RT in either group. RT did not have negative impact on body mass index (BMI), weight, NRS, CRP, Alb, Prealb, or PNI. Overall survival (OS) and disease-free survival (DFS) were significantly better in patients with a higher BMI in the HPV- group (OS, *p* = 0.011; DFS, *p* = 0.028); DFS was significantly better in patients with C-reactive protein (CRP) < 3.5 g/dL in the HPV- (*p* = 0.021) and HPV+ (*p* = 0.018) groups, and with total lymphocyte count (TLC) >1.28/mm^3^ in the HPV+ group (*p* = 0.014). Higher NRS 2002 was an independent adverse prognostic factor for OS and DFS in HPV-, but not in the HPV+ group. Kaplan–Meier analysis showed that both OS and DFS were significantly better in HPV- patients with lower NRS 2002 scores. However, this relationship was not observed in the HPV+ group. Conclusions: Regardless of HPV status, patients with OPC can develop malnutrition during RT/CRT. Therefore, nutritional support during RT/CRT is required in patients with HPV- and HPV+ OPC.

## 1. Introduction

The incidence of oropharyngeal carcinoma (OPC) has been increasing recently, mostly in young patients, which could be associated with a relatively new risk factor—infection with the high-risk human papilloma virus (HPV). HPV infection is usually not associated with other well-known typical determinants of OPC such as smoking, alcohol abuse, diet, chemical irritants, or poor oral hygiene [1,2]. Typical patients with HPV-related OPC are younger men with a high social status, low rate of comorbidities, and a history of many sexual partners and oral–genital sexual practice. Patients with HPV-related OPC have a better prognosis and longer survival compared to patients with HPV- OPC with typical risk factors (e.g., smoking, alcohol abuse) [3,4]. A better prognosis is also observed in HPV+ patients with more advanced OPC with lymph node involvement [3,4]. HPV+ OPC is more responsive to radiotherapy (RT) and chemoradiotherapy (CRT) [5]. Due to the abovementioned differences between HPV-positive and -negative OPC patients, the current eighth edition of the American Joint Committee on Cancer (AJCC) Staging Manual reflects HPV infection status in determining the clinical stage of OPC. The eighth edition of the AJCC Staging Manual presents p16 immunohistochemistry findings as a marker of HPV infection [6,7].

Malnutrition is a common problem in OPC patients. It is caused by insufficient food intake due to dysphagia, odynophagia, and lack of appetite caused by the tumor. It is also secondary to RT/CRT as a consequence of mucositis, with dry mouth, loss of taste, and dysphagia. The severe dysphagia is the most serious challenge in patients with OPC, because in 20–30% of patients it leads to the definitive, total impossibility of eating through the mouth. These patients require permanent percutaneous gastrostomy tubes [1]. Although there are many reports concerning the deterioration of nutritional status (NS) and nutritional intervention in patients with head and neck squamous-cell carcinoma (HNSCC) undergoing RT or CRT [8,9,10,11], there is lack of information regarding differences in NS between patients with HPV- and HPV+ OPC. This knowledge can be helpful in the management of these patients. Proper assessment of NS enables the appropriate nutritional therapy in order to support the care of OPC patients.

The aim of this study was to assess and compare the NS of patients with HPV+ and HPV- OPC before and after RT or CRT, using various anthropometric, clinical, and biochemical parameters.

## 2. Materials and Methods

### 2.1. Patients

The analysis included 127 patients with OPC who received definitive radical RT/CRT at the I Radiation and Clinical Oncology Department of Maria Skłodowska-Curie Research Institute of Oncology, Gliwice Branch, Poland, in the period 2012–2016. Assessment of the NS was performed in patients before and after treatment (RT or CRT). There were 87 (68%) men and 40 (32%) women in the analyzed group, with a mean age of 60.5 years (range: 30–80 years). The following inclusion criteria were used: primary OPC, age > 18 years, regionally advanced cancer without distant metastases (T1–T4, N0–N3, M0). Exclusion criteria included surgical treatment, cancer recurrence, and incomplete demographic and/or clinical data.

### 2.2. Funding Statement and Ethics Approval and Consent to Participate

This study was supported by a grant from the National Centre of Research and Development, Poland (grant TANGO2/340829/NCBR/2017, A.M. Mazurek).

All procedures performed in studies involving human participants were in accordance with the ethical standards of the institutional research committee (the Bioethics Committee at Maria Skłodowska-Curie Research Institute of Oncology, Gliwice Branch, KB/430-18/13), and with the 1964 Helsinki Declaration and its later amendments or comparable ethical standards. Informed consent was obtained from all individual participants included in the study.

### 2.3. Study Design

The patients were radically irradiated with the total dose, sterilizing the squamous-cell carcinoma tissue. Most of the patients received the conventionally fractionated 70 Gy dose. Accelerated schemes were also used, including continuous accelerated irradiation (CAIR) and simultaneous integrated boost (SIB). Therefore, the dose range was in the 66–72 Gy range. Despite the obvious differences in the total physical dose, it can be assumed that the total biological doses were not different. Simultaneously, during radiotherapy, cisplatin was administered at a dose of 100 mg/m^2^ on irradiation days 1, 22, and 43, or at a dose of 40 mg/m^2^ administered weekly. In the case of induction chemotherapy, 2–3 cycles were used according to the PF regimen (cisplatin and 5-fluoruracil) or the TPF regimen (docetaxel, cisplatin, and 5-fluorouracil).

All patients were asked about deterioration of NS, body weight before disease and before treatment, loss of body weight, and food intake since the onset of disease. Information on smoking (including the amount and duration of smoking and/or smoking cessation after diagnosis) and alcohol consumption was collected. The patients’ height and weight were measured, and laboratory blood tests were performed before and after treatment. The selected blood count parameters and biochemical parameters (i.e., albumin, prealbumin, total lymphocyte count (TLC), hemoglobin, C-reactive protein) were analyzed. The body mass index (BMI) and weight loss over the course of the disease were calculated. Percentage weight change during treatment was calculated using weight before RT as a baseline and weight after the end of RT. The patients were divided into two groups according to their BMI: malnourished patients (BMI < 18.5 kg/m^2^), and well-nourished patients (BMI ≥ 18.5 kg/m^2^) [12]. The nutritional risk according to NRS 2002 (Nutritional Risk Score 2002) by the European Society of Parenteral and Enteral Nutrition (European Society of Parenteral and Enteral Nutrition, ESPEN) was assessed [13,14]. Onodera’s prognostic nutritional index (PNI) was calculated based on the serum albumin concentration and total lymphocyte count in the peripheral blood, using the following formula: 10 × level of albumin (g/dL) + 0.005 × total lymphocyte count (/mm^3^) [15].

The patients were divided into two groups according to their HPV involvement: HPV-related and non-HPV-related groups. Confirmation of HPV etiology was obtained from FFPE (formalin-fixed, paraffin-embedded) tissue material and/or extracellular circulating HPV DNA.

The stage of OPC was classified according to the eighth edition of the American Joint Committee on Cancer (AJCC)’s TNM classification system: primary tumor (T), regional lymph node metastasis (N), or distant metastasis (M) [6,7]. Tumor diameter and lymph node invasion were assessed on the basis of computed tomography (CT) of the head and neck region.

The clinical, anthropometric, and laboratory parameters before and after treatment were compared between HPV- and HPV+ patients. The parameters before treatment were signified as “0”, and after treatment as “1”.

The median follow-up was 74.58 (0.1–165.58) months. Overall survival (OS) and disease-free survival (DFS) were analyzed in both groups. OS was defined as the time from randomization to death from any cause. DFS was defined as the time from randomization to the first event of either recurrent disease or death.

Comparisons between all HPV- and HPV+ patients, as well as between HPV- and HPV+ patients undergoing RT and CRT separately, were performed to exclude the concomitant chemotherapy as a confounding factor in the analysis of the whole cohort. HPV+ OPC patients were treated more predominantly with CRT than HPV- OPC patients.

### 2.4. Confirmation of the HPV Etiology

Confirmation of the HPV etiology was carried out before treatment using FFPE tissue material (if it was available) and/or extracellular circulating HPV DNA (standard analysis in all OPC patients in our institute).

#### 2.4.1. Tissue Material

Formalin-fixed, paraffin-embedded tumor samples were examined for high-risk HPV (HR-HPV) infection using a double-check algorithm, including immunohistochemical assessment of p16 (INK4A) protein expression, followed by the detection of HR-HPV DNA in tumor tissue using real-time PCR. Only cases with both p16INK4A expression and HR-HPV DNA amplification were classified as truly HR-HPV-positive.

#### 2.4.2. Analysis of cfHPV16 DNA in Plasma

Peripheral blood (12 mL) was collected in K3EDTA tubes (Becton–Dickinson, Franklin Lakes, NJ, USA). Plasma was separated within an hour by double centrifugation at 300× *g* and 1000× *g*, both at 4 °C for 10 min. DNA was extracted (according to the manufacturer’s instructions) from 1 mL of plasma using the Genomic Mini AX Body Fluids Kit (A&A Biotechnology, Gdynia, Poland). Each measurement consisted of a standard curve of three dilutions of a plasmid construct containing the HPV16 genome, a negative control, and a sample. For HPV16 detection, a reaction was performed using primers and a probe set for the HPV16 genome. PCR reactions were performed using the Bio-Rad CFX96 qPCR instrument (Bio-Rad Laboratories, Hemel Hempstead, UK). If HPV16 was found, its presence was confirmed with a second independent DNA isolation.

### 2.5. Statistical Analysis

The categorical variables were presented as numbers and percentages. Continuous variables with normal distribution were expressed as the means and standard deviations. The Shapiro–Wilk test was used to determine statistical distribution in the analyzed patients. The Mann–Whitney *U* test was used to compare the HPV+ and HPV- groups. The Wilcoxon test was used to compare pre- and post-treatment parameters in all patients and both HPV groups separately. Multiple testing correction was performed using Bonferroni correction. Cox regression analysis was used to calculate overall survival (OS) and disease-free survival (DFS). The log-rank test was used to assess the equality of survival distributions across different strata. Receiver operating characteristic (ROC) curve analysis was performed to determine the optimal cutoff values for prognostic factors related to DFS and OS. Youden’s index was selected as the approximate cutoff value for each parameter. Kaplan–Meyer curves were constructed to determine the impact of selected nutritional parameters on OS and DFS in HPV- and HPV+ OPC patients. A *p*-value of less than 0.05 was considered to be statistically significant. The statistical analyses were performed using the Statistica^®^ software program, version 13.0: Dell Inc. (2016). Dell Statistica (data analysis software system), version 13. software.dell.com. (StatSoft Poland, Kraków, Poland).

## 3. Results

### 3.1. General Characteristics

The general clinical characteristics of the 127 patients are presented in Table 1. The clinicopathological features and basic laboratory results of all patients and in both the HPV- and HPV+ groups, before and after treatment, are presented and compared in Table 1, Table 2, Table 3 and Table 4 and Appendix A.

Initial BMI (before treatment) was significantly lower in the HPV- group (72.38 ± 15.24 (44–104.80) vs. 79.41 ± 14.64 (46–111) kg/m^2^; *p* = 0.009). Final BMI (after treatment) was similar in both groups (*p* = 0.419).

Smoking and alcohol consumption were significantly more frequently reported in HPV- patients compared to HPV- patients (75% vs. 39%; *p* = 0.004, and 78.1% vs. 55.9%; *p* = 0.012, respectively). The compared groups were different with regard to the detailed tumor locations, histopathological grading, and lymph node invasion. The tonsil location was more frequently observed in HPV- patients, and palate locations were noted only in HPV- patients (*p* = 0.010). G3 grading was significantly more frequent in HPV+ patients compared to HPV- patients (*p* = 0.049). Lymph node invasion was significantly more common in HPV+ patients compared to HPV- patients (*p* = 0.029) (Table 1).

In clinical manifestation, the rates of dysphagia (19 (27.9%) vs. 15 (25.40%); *p* = 0.906) and weight loss (5 (7.40%) vs. 3 (5.10%); *p* = 0.724) were comparable in both groups (HPV- and HPV+ groups, respectively). The analyzed groups were different in terms of the rate of neck tumor, which was significantly more frequent in HPV+ patients (29 (49.20%)) compared to the HPV- group (17 (25.00%)) (*p* = 0.008) (Table 1).

The duration of medical history was similar in HPV- and HPV+ patients (5.92 ± 60 vs. 5.13 ± 7.26 months; *p* = 0.403). The treatment duration was significantly longer in HPV+ patients compared to HPV- patients, which was associated with more frequent use of CRT in these patients (48.51 ± 10.04 vs. 43.64 ± 9.00; *p* = 0.005).

### 3.2. Correlation between Local Tumor Stage and Treatment Regimen in HPV- and HPV+ Patients

The correlation between local tumor stage and treatment type was determined. There was a significantly higher number of more advanced N2–3 tumors in HPV+ patients compared to HPV- patients. In the comparison of cancer stage for the RT and CRT groups separately, there was no differences in lymph node invasion between the HPV- and HPV+ groups. RT was significantly more frequent in N0–1 tumors, whereas CRT was more common in N2–3 tumors, in the analysis of all patients together. In comparisons in the HPV- and HPV+ groups separately, this significant difference was noted only for HPV- patients (Table 2).

### 3.3. Comparison of Nutritional Parameters before and after Treatment in HPV- and HPV+ Patients

The baseline weight (M0) was significantly higher in HPV+ OPC patients compared to those with HPV- OPC (79.41 ± 14.64 (46–111) vs. 72.38 ± 15.24 (44–104.80); *p* = 0.009). There was no statistical difference in terms of weight after treatment between the two groups. The weight after treatment (M1) was also higher in HPV+ patients, but not significantly (72.96 ± 14.81 (45–110) vs. 70.56 ± 15.24 (42–94.70); *p* = 0.419) (Table 1). Loss of weight (LOW) during CRT was similar in HPV- and HPV+ patients: −5.65 ± 4.85 (−19.35–4.84) vs. −7.19 ± 5.19 (−20.56–1.85); *p* = 0.144, *p* H–B = 1.000 (Table 3). It should be noted that in both groups, the mean LOW was critical (≥5%). Critical weight loss (CWL) was noted in 23 (54.8%) HPV- and 29 (65.9%) HPV+ patients (*p* = 1.00).

Post-treatment evaluations were performed depending on the treatment duration reported in Table 1. The treatment duration was significantly longer in HPV+ patients, due to the more frequent use of CRT in this group of patients.

Analogical observation was used to determine the nutritional risk according to the NRS 2002 in OPC patients. Baseline values of NRS 2002 were similar in HPV- and HPV+ patients (0.41 ± 0.87 (0–4) vs. 0.28 ± 0.64 (0–4); *p* = 0.680). After treatment, NRS 2002 was not significantly higher in HPV+ patients (3.39 ± 1.10 (1–5) vs. 2.98 ± 1.09 (1–5); *p* = 0.074). This difference was nearly statistically significant. Thus, there was a clear trend of dependency (Table 1). Comparing NRS 2002 before and after treatment in both groups separately, a significant increase in NRS 2002 scores after RT/CRT was reported in the two groups (*p* < 0.0001, *p* H–B < 0.0001) (Table 4).

In addition, the distribution of the NRS 2002 groups (<3 vs. ≥3) significantly changed after treatment. Before treatment, most patients (97.6% and 98.0%) were classified as NRS 2002 < 3, and significantly fewer patients (2.4% and 2.0%) had NRS 2002 ≥ 3 (i.e., were at nutritional risk). These proportions changed significantly after treatment, as follows: NRS 2002 < 3 was reported in 35.7% and 25.5%, while NRS 2002 ≥ 3 was noted in 64.3% and 74.5% of HPV- and HPV+ patients, respectively (*p* < 0.0001). Thus, the percentage of patients with nutritional risk increased significantly after treatment in both groups. There was no difference in the proportion of HPV- and HPV+ patients in each of the NRS 2002 groups before and after treatment (*p* = 0.62 and *p* = 0.36, respectively).

Most laboratory results were comparable in both groups. TLC 1 was significantly lower in HPV+ compared to HPV- patients (*p* = 0.004, *p* H–B = 0.048). C-reactive protein (CRP) 0 was not significantly higher in HPV- than in HPV+ patients (*p* = 0.008, *p* H–B = 0.088). CRP 1 was comparable in both groups (*p* = 0.179). A comparable CRP increase was noted after treatment in both groups (*p* = 0.431). The correlations between TLC and CRP before and after treatment were analyzed and excluded: for TLC 0 and CRP 0: R = 0.053 (*p* = 0.563); for TLC 1 and CRP 1: R = −0.116 (*p* = 0.214), using Spearman’s test.

PNI was comparable in both groups before and after treatment. A comparable decrease in PNI after treatment was noted in both groups. The differences in laboratory parameters before and after treatment in both groups are presented in Appendix A.

### 3.4. Comparison of Nutritional Parameters before and after Treatment in HPV- and HPV+ Patients Undergoing RT and CRT Separately

Due to the different treatment regimens (more frequent CRT in HPV+ patients), additional comparisons of nutritional parameters before and after treatment in patients undergoing RT and CRT separately were performed (Table 5, Appendix A).

Differences in weight, BMI, laboratory results (i.e., CRP, albumin, prealbumin, hemoglobin, TLC), and PNI before and after treatment were similar in the HPV- and HPV+ groups.

CRT caused significant weight loss, decreases in BMI, albumin, TLC, hemoglobin concentration, and increases in NRS 2002 score in HPV- and HPV+ patients. A significant decrease in prealbumin levels after CRT was noted only in HPV+ patients. RT caused a significant decrease in hemoglobin concentration and TLC in HPV- patients. There were no significant differences regarding other parameters after RT in either group. RT did not have a negative impact on BMI, weight, NRS, CRP, Alb, Prealb, or PNI.

Comparison of differences in weight, BMI, NRS, PNI, and laboratory results before and after CRT and RT between HPV- and HPV+ patients did not show any significant differences between groups according to HPV status.

### 3.5. Comparison of Selected Clinicopathological Factors between Groups with Low and High Prognostic Nutritional Index (PNI)

The patients were divided into two groups according to the cutoff value of the mean PNI. The mean PNI values (41.68 for all patients, 41.59 for HPV- patients, and 41.78 for HPV+ patients) among the study population were set as the border values to divide the high- and low-PNI groups in order to perform statistical comparisons of clinicopathological findings between both groups. Comparison using Holm–Bonferroni correction did not show any statistically significant differences between the low- and high-PNI subgroups in both HPV- and HPV+ patients (*p* > 0.05) (Appendix A).

### 3.6. OS and DFS in HPV- and HPV+ Patients

#### 3.6.1. Kaplan–Meier Analysis

OS and DFS in all patients and both HPV groups are presented in Figure 1A,B and Figure 2A,B, respectively. OS (*p* = 0.011) and DFS (*p* = 0.028) were significantly better in HPV- patients with higher BMI. OS was comparable regardless of TLC in the HPV- (*p* = 0.294) and HPV+ (*p* = 0.501) groups. DFS in the whole cohort (*p* = 0.026) and DFS in the HPV+ group (*p* = 0.014) were better in patients with a higher TLC > 1.28. A similar, but non-significant result was noted in the HPV- group (*p* = 0.262). There was no clear impact of the hemoglobin (HB) level on OS in the HPV- (*p* = 0.337) and HPV+ (*p* = 0.661) groups. There was better DFS in patients with HB > 13.5 g/dL in the HPV- group (*p* = 0.017). There was better OS in patients with CRP < 3.50 g/dL in the whole cohort (*p* = 0.008). DFS was significantly better in patients with CRP < 3.5 g/dL in both groups (*p* = 0.018 HPV+, *p* = 0.021 HPV-). There was no influence of serum albumin levels on OS (*p* = 0.100 HPV+, *p* = 0.751 HPV-) and DFS (*p* = 0.952 HPV+, *p* = 0.807 HPV-) in either group. PNI did not influence OS in either group (*p* = 0.561 HPV+, *p* = 0.932 HPV-). There was significantly better DFS in patients with a higher PNI (>39) in the whole cohort (*p* = 0.042), and a non-significant increase for both HPV groups separately (*p* = 0.275 HPV+, *p* = 0.146 HPV-). There was no statistical difference between HPV- and HPV+ patients in terms of OS and DFS according to loss of weight (<5% vs. >5%) in either group (*p* > 0.05) (Appendix A). Regarding NRS 2002, there were no statistically significant differences in OS and DFS when the patients were divided into two subgroups (NRS 2002 < 3 vs. NRS 2002 ≥ 3) in either HPV- or HPV+ patients (*p* > 0.05) (Appendix A). Significantly worse OS and DFS were noted in HPV- patients (but not in HPV+ patients) in the division NRS 2002 < 2 vs. NRS 2002 ≥ 2 (Figure 3).

#### 3.6.2. Prognostic Factors for Survival in Cox Regression Analysis in HPV- and HPV+ Patients

##### Prognostic Factors for OS

In HPV- patients, higher pre-treatment CRP (HR = 3.45; 95% CI: 1.37–8.68; *p* = 0.008), NRS 2002 (HR = 3.83; 95% CI: 1.22–12.01; *p* = 0.021), and alcohol abuse (HR = 6.39; 95% CI: 1.40–29.14; *p* = 0.017) were significant independent adverse prognostic factors, while female gender (HR = 0.12; 95% CI: 0.02–0.58; *p* = 0.009) was a good predictor for OS. There were no significant prognostic factors for OS in HPV+ patients (Table 6).

##### Prognostic Factors for DFS

In HPV- patients, higher pre-treatment CRP (HR = 2.90; 95% CI: 1.08–7.76; *p* = 0.034), higher lymph node status (HR = −4.26; 95% CI: 1.59–11.43; *p* = 0.004), higher NRS 2002 (HR = 5.89; 95% CI: 1.79–19.37; *p* = 0.003), and alcohol abuse (HR = 8.01; 95% CI: 1.41–45.39; *p* = 0.0019) were significant independent adverse prognostic factors for DFS, while higher hemoglobin concentration (HR = 0.34; 95% CI: 0.13–0.90; *p* = 0.029) was a good predictor for DFS. In HPV+ patients, only higher TLC (HR = 0.20; 95% CI: 0.05–0.82; *p* = 0.025) was a good, significant, independent prognostic factor for DFS (Table 7).

#### 3.6.3. Prognostic Factors for Survival in Cox Regression Analysis in the Whole Cohort (without Division into Groups According to HPV Status)

Higher pre-treatment CRP (HR = 2.22; 95% CI: 1.12–4.42; *p* = 0.023), tonsil location (HR = 3.20; 95% CI: 1.19–8.60; *p* = 0.021), and alcohol abuse (HR = 3.86; 95% CI: 1.00–14.86; *p* = 0.050) were independent adverse prognostic factors, while higher BMI (HR = 0.32; 95% CI: 0.14–0.72; *p* = 0.006) and positive HPV status (HR = 0.36; 95% CI: 0.16–0.80; *p* = 0.012) were good prognostic factors for OS (Appendix A).

Higher pre-treatment CRP (HR = 3.53; 95% CI: 1.50–8.32; *p* = 0.004), higher lymph node status (HR = 3.28; 95% CI: 1.33–8.12; *p* = 0.010), and alcohol abuse (HR = 5.78; 95% CI: 1.04–31.97; *p* = 0.045) were independent adverse prognostic factors for DFS. Positive HPV status (HR = 0.33; 95% CI: 0.14–0.81; *p* = 0.015) was a good prognostic factor for DFS (Appendix A).

## 4. Discussion

Recently, a novel HPV-related type of OPC has been described. Patients suffering from HPV-related OPC are different from HPV- patients. Typical HPV- OPC is associated with smoking and alcohol abuse. Patients with HPV- OPC are older and less healthy (with comorbidities) at diagnosis compared to HPV+ patients [16,17,18]. Grohoj et al. observed more frequent comorbidities in HPV- patients compared to HPV+ patients [19]. Authors have observed cerebrovascular disease, peripheral vascular disease, dementia, ulcer disease, and liver disease most often in HPV- OPC patients. The higher risk of multiple comorbidities in HPV- patients may be associated with their older age, smoking, and alcohol consumption. Our results show significantly less frequent smoking and alcohol abuse in HPV+ OPC patients [19].

Patients with HPV+ OPC usually present with a more advanced stage of disease, but in spite of this their prognosis is better [20]. This was confirmed in our study, where HPV+ OPC patients had more advanced lymph node status with predominating G3 grading. In our study, OS and DFS were also significantly better in HPV+ compared to HPV- OPC patients.

Malnutrition is an important problem in patients with HNSCC, including OPC, and is associated with aggressive disease located in the oropharynx and severe side effects related to the treatment. Predominant acute side effects—such as xerostomia, mucositis, loss of appetite, odynophagia, dysphagia, nausea, and vomiting—usually lead to weight loss and dehydration. Moreover, many publications have proven that RT also decreases NS in these patients. In such patients, nutritional intervention is needed. Most frequently, a feeding tube (gastrostomy/jejunostomy) is used in order to feed these patients [21,22,23,24,25,26,27,28]. Greater nutritional impairment is observed after CRT compared to RT [18]. Our study confirms nutritional impairment following RT/CRT, because all nutritional parameters decreased significantly after treatment in both groups of patients. Vangelov et al. [26,28] noted that CRT and HPV+ status were predictive of critical weight loss. In our study, a greater decrease in most nutritional parameters was also observed in HPV+ compared to HPV- patients, which was associated with the more frequent CRT in the former group. RT alone, as the only treatment, was more common in HPV- patients, mostly due to them being at less advanced stages of the disease. However, in the comparison of the NS of HPV- and HPV+ OPC patients following RT and CRT separately, this difference was not found. CRT adversely affected the NS of patients regardless of their HPV status. RT caused a significant decrease in hemoglobin concentration and TLC in HPV- patients. There were no significant differences regarding other nutritional parameters after RT in either group. Generally, RT/CRT worsened the NS of patients with OPC, regardless of their HPV status, because in both the HPV- and HPV+ groups there was a significant decrease in all analyzed nutritional parameters after treatment. Differences in weight, BMI, laboratory results, and PNI before and after RT and CRT were similar in the HPV- and HPV+ groups. Therefore, deterioration of NS after treatment was similar in HPV- and HPV+ patients. Therefore, our study shows that HPV+ status associated with more frequent CRT (but not HPV status alone) predicts the greater deterioration of NS in OPC patients. Some authors have reported that HPV+ OPC patients have higher acute toxicity compared to HPV- patients, and dysphagia is also more frequent in these patients [21,22,23]. In our study, deterioration of NS in HPV- and HPV+ patients was similar, and was greater after CRT compared to RT. Thus, our findings confirm those of the abovementioned studies [18,21,22,23]. Our study is one of the first publications regarding the influence of HPV status on NS in patients with OPC before and after RT/CRT. To our knowledge, it is also the first study analyzing the impact of nutritional parameters on survival according to HPV status. Therefore, there are not a lot of studies for comparison of our results with data from the literature.

Vangelov et al. [26,28] noted that HPV+ OPC patients were at higher nutritional risk compared to the rest of the cohort in their study. The authors concluded that HPV+ patients had undergone significantly greater weight loss following RT compared to the rest of their cohort (8.4% vs. 6.1%, *p* = 0.003). Critical weight loss (>5%) was noted in 86% of patients, with a higher percentage in the HPV+ group (*p* = 0.011). The authors noted that the mean percentage weight change in both HPV status groups in their study was at a critical level (≥5%) [28]. Our findings do not totally confirm the aforementioned results, because although the mean loss of weight was also higher in HPV+ patients (7.19% (HPV+) vs. 5.65% (HPV-)), this difference was not statistically significant. The percentage of patients achieving critical weight loss was comparable in our cohort (54.8% vs. 65.9% for the HPV- and HPV+ groups, respectively).

In our study, OS and DFS were significantly longer in patients with higher BMI in all patients and in the HPV- group, and not significantly for the HPV+ group. The association between higher BMI and longer survival was presented in the literature. According to Ottosson et al. [29], higher BMI (>25 kg/m^2^) before RT is positively associated with survival in patients with OPC. Moon et al. [30] reported that BMI < 18.5 kg/m^2^ was an independent predictor of cancer-specific survival (CSS) and OS in HNSCC patients who had undergone definitive CRT.

The importance of HPV status as a prognostic marker in OPC patients has been well established in the literature [16,17,18]. The literature’s data indicate that tumor HPV status is a strong and consistent determinant of superior survival, regardless of treatment strategy (surgery, RT, concurrent CRT, or induction chemotherapy plus concurrent CRT), with 5-year survival rates among patients with HPV+ OPC of approximately 75% to 80%, vs. 45% to 50% among patients with HPV- OPC. The superior OS in HPV+ OPC patients is associated with a better response to oncological treatment (CRT) [18,31,32,33,34,35]. It has been also reported that concurrent chemotherapy improves survival rates in patients with HNSCC compared with RT alone, but frequently at the cost of increased rates of mucositis and dysphagia [36].

According to the literature, age, smoking, tumor stage, and treatment also have an important influence on survival in patients with OPC [2]. Our study found that RT/CRT had an adverse effect on the NS of OPC patients regardless of their HPV status. The baseline weight and BMI before treatment were significantly higher in the HPV+ group, and they were comparable in both groups after treatment. Weight and BMI after treatment were also higher in the HPV+ group, but not significantly. Weight and BMI decreased after treatment in both groups. CWL was reported in both groups. Only total lymphocyte count (TLC) after treatment was significantly lower in HPV+ compared to HPV- patients, because there was a greater TLC decrease in the HPV+ group compared to the HPV- group. PNI was comparable in both groups before and after treatment. Decreased PNI was noted in both groups after treatment, and was comparable in both groups. Generally, similar decreases in nutritional parameters were reported in HPV- and HPV+ patients, confirming the results of previous studies [37]. Other studies have shown the greater deterioration of NS in HPV+ patients compared to HPV- patients [20,21].

It should be added that our study did not show any clear impact of nutritional status (including PNI) on survival in patients with OPC, regardless of HPV infection. This may be associated with other, stronger factors—including HPV status—that determine survival in patients with OPC. It is possible that positive HPV status has a stronger influence on survival than nutritional parameters. This hypothesis is supported by the fact that our study found a greater association between NS and survival in HPV- patients. Higher NRS 2002 was an independent adverse prognostic factor for OS and DFS in HPV- patients, but not in the HPV+ group. The significant impact of NRS 2002 on survival in HPV- patients was confirmed via Kaplan–Meier analysis, where both OS and DFS were significantly better in HPV- patients with lower NRS 2002 scores. This relationship was not observed in the HPV+ group. The impact of NRS 2002 on survival according to HPV status in OPC patients has been not reported in the worldwide literature. According to some authors, a higher PNI predicts better survival in patients with HNSCC [38,39,40,41]. Our study did not confirm these reports.

Our study confirmed that HPV+ OPC patients had better NS at diagnosis compared to those with HPV- OPC [28,41,42,43]. This is associated with the fact that HPV+ OPC patients are generally younger and healthier (i.e., with fewer comorbidities) compared to HPV- OPC patients. Studies comparing the prevalence of malnutrition in HPV+ and HPV- patients after treatment report similar, if not worse, nutritional outcomes in HPV+ patients compared to HPV- patients [26,40,41]. This suggests that HPV+ OPC patients are susceptible to a greater deterioration of NS, and are potentially at greater risk of treatment-related malnutrition than HPV- patients. Therefore, they need careful consideration and nutritional support [42,43]. In a study by Harrowfield et al. [37], deterioration of NS was similar in HPV- and HPV+ patients after treatment, and patients required similarly intense nutritional intervention. Similar to this research, in our study, deterioration of NS did not depend on HPV status, and was comparable in both groups. Thus, the results of previously published studies are contradictory. Therefore, further investigations regarding the impact of HPV status on NS in HPV- and HPV+ patients are required. This knowledge is very important, and should be explored, because the incidence of OPC associated with HPV infection has significantly increased in the last few decades, and the overall incidence of HPV- OPC has decreased, probably due to the reduction in smoking in the population [44].

The strength of this study is that it tackles a clinically relevant problem regarding the changes in nutritional status during CRT, and their impact on survival in OPC patients depending on HPV status. There are no similar articles in the global literature. To the best of our knowledge, this is the first study to conduct a comprehensive analysis of NS, including its impact on survival, in HPV- and HPV+ OPC patients. This study was conducted in a large tertiary oncological center with experienced, well-established, multidisciplinary teams.

The single-center observation and retrospective analysis are limitations of this study. A prospective randomized multicenter study is needed to assess the influence of NS on survival in HPV- and HPV+ patients with OPC. To our knowledge, this research is one of the first studies in the worldwide literature to conduct a comprehensive comparative analysis of the nutritional status of HPV- and HPV+ OPC patients.

## 5. Conclusions

Despite differences in local stage and treatment (HPV+ cancers were more advanced in terms of lymph node invasion, and were more frequently treated using CRT), NS was comparable in both groups. CRT adversely affected the NS of patients regardless of their HPV status. RT caused significant decreases in hemoglobin concentration and TLC in HPV- patients. There were no significant differences regarding other nutritional parameters after RT in either group. Generally, RT/CRT worsened the NS of patients with OPC regardless of their HPV status, because in both the HPV- and HPV+ groups, there were significant decreases in all analyzed nutritional parameters after treatment. Differences in weight, BMI, laboratory results (i.e., CRP, albumin, prealbumin, hemoglobin, TLC), and PNI before and after RT and CRT were similar in the HPV- and HPV+ groups. Therefore, deterioration of NS after treatment was similar in HPV- and HPV+ patients. A greater association between NS and survival was noted in HPV- patients. This might be associated with a fact that HPV+ status is a stronger predictive factor for survival than nutritional parameters. Higher NRS 2002 was an independent adverse prognostic factor for OS and DFS in the HPV- patients, but not in the HPV+ group. The significant impact of NRS 2002 on survival in HPV- patients was confirmed via Kaplan–Meier analysis, where both OS and DFS were found to be significantly better in HPV- patients with lower NRS 2002 scores. However, this relationship was not observed in the HPV+ group. OS and DFS were significantly better in patients with NRS 2002 < 2 compared to NRS 2002 ≥ 2 for HPV- patients, and not significantly for the HPV+ group.

Our study showed that both HPV- and HPV+ OPC patients can develop malnutrition during RT/CRT. Therefore, regardless of HPV status, nutritional support is required for OPC patients during RT/CRT. Further prospective multicenter observational studies regarding the impact of HPV status on NS and the influence of nutritional parameters on survival in OPC patients are needed.

## Figures and Tables

**Figure 1 cancers-14-02335-f001:**
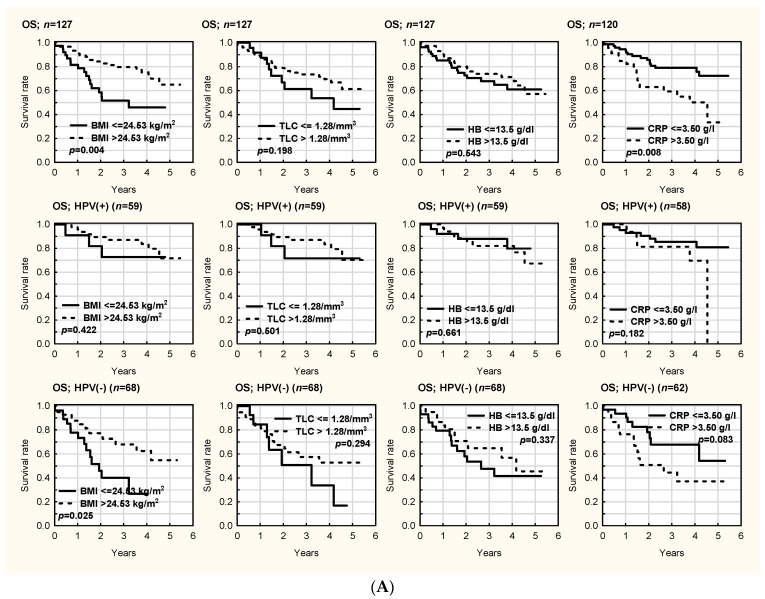
(**A**). Overall survival (OS) in HPV-, HPV+, and all patients according to body mass index (BMI), total lymphocyte count (TLC), hemoglobin (HB), and C-reactive protein (CRP) levels. (**B**). Overall survival (OS) in HPV-, HPV+, and all patients according to albumin and prealbumin levels and prognostic nutritional index (PNI).

**Figure 2 cancers-14-02335-f002:**
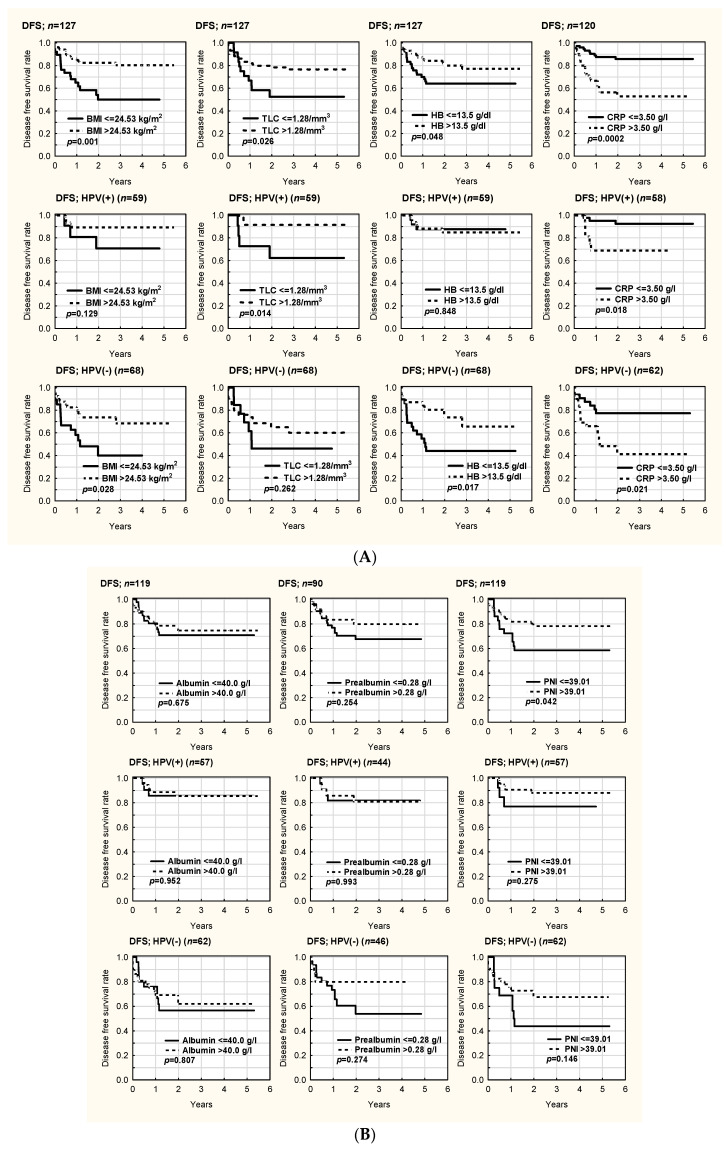
(**A**). Disease-free survival (DFS) in HPV-, HPV+, and all patients according to body mass index (BMI), total lymphocyte count (TLC), hemoglobin (HB) level, and C-reactive protein (CRP) level. (**B**). Disease-free survival (DFS) in HPV-, HPV+, and all patients according to albumin and prealbumin levels and prognostic nutritional index (PNI).

**Figure 3 cancers-14-02335-f003:**
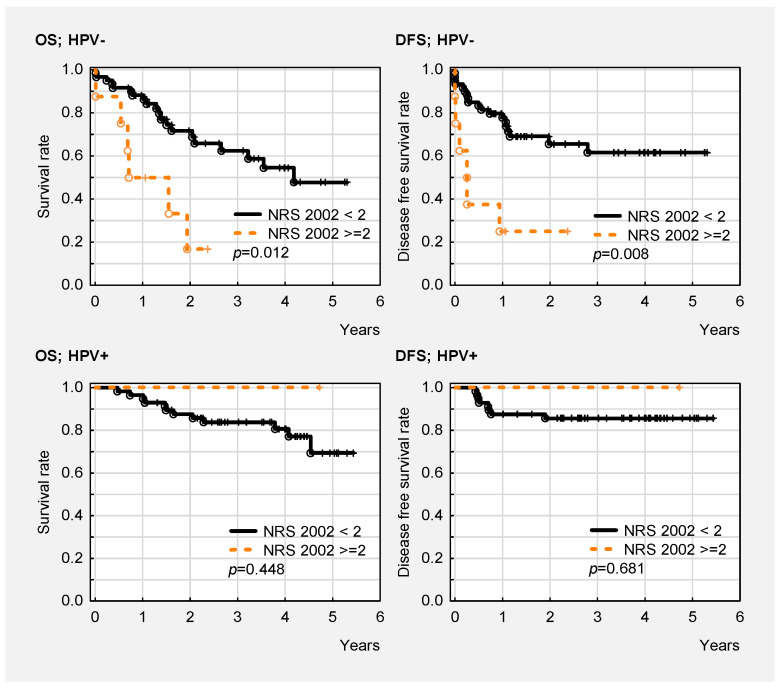
Overall and disease-free survival in HPV- and HPV+ patients according to NRS 2002 (NRS 2002 < 2 vs. NRS 2002 ≥ 2).

**Table 1 cancers-14-02335-t001:** The patients’ general clinicopathological characteristics and basic laboratory results.

Feature	All	HPV(-)	HPV(+)	*p*
Demographic characteristics				
Age (years)	60.62 ± 8.54 (30–80)	60.85 ± 7.48 (37–79)	60.36 ± 9.67 (30–80)	0.745
Male/female	87 (68.5%)/40 (31.5%)	51 (75.1%)/17 (25.0%)	36 (61.0%)/23 (39.0%)	0.133
Weight (kg)				
Before treatment	75.65 ± 15.24 (44–111)	72.38 ± 15.24 (44–104.80)	79.41 ± 14.64 (46–111)	0.009
After treatment	71.88 ± 14.17 (42–110)	70.56 ± 15.24 (42–94.70)	72.96 ± 14.81 (45–110)	0.419
Weight difference (kg)	−5.08 ± 4.04 (−17.20–3.00)	−4.32 ± 3.71 (−12.00–3.00)	−5.71 ± 4.23 (−17.20–1.00)	0.099
Weight difference (%)	−6.49 ± 5.07 (−20.56–4.84)	−5.65 ± 4.85 (−19.35–4.84)	−7.19 ± 5.19 (−20.56–1.85)	0.144
BMI (kg/m^2^) groups				0.371
<18.5	5 (3.90%)	4 (5.90%)	1 (1.70%)	
>18.5	122 (96.10%)	64 (94.10%)	58 (98.30%)	
BMI (kg/m^2^) groups				0.232
<20	10 (7.90%)	9 (13.20%)	1 (1.70%)	
20–25	39 (30.70%)	24 (35.30%)	15 (25.40%)	
25–30	51 (40.20%)	21 (30.90%)	30 (50.80%)	
>30	27 (21.30%)	14 (20.60%)	13 (22.00%)	
NRS 2002				
Before treatment	0.35 ± 0.77 (0–4)	0.41 ± 0.87 (0–4)	0.28 ± 0.64 (0–4)	0.680
After treatment	3.20 ± 1.11 (1–5)	2.98 ± 1.09 (1–5)	3.39 ± 1.10 (1–5)	0.074
Smoking				0.0004
No	35 (27.60%)	8 (11.80%)	27 (45.80%)	
Yes	74 (58.30%)	51 (75.00%)	23 (39.00%)	
Smoking cessation	18 (14.20%)	9 (13.20%)	9 (15.30%)	
Alcohol				0.012
No	41 (32.30%)	15 (22.10%)	26 (44.10%)	
Normal drinking	83 (65.40%)	50 (73.50%)	33 (55.90%)	
Alcohol abuse	3 (2.40%)	3 (4.40%)	0 (0.00%)	
Detailed tumor location				0.010
Tonsil	91 (71.70%)	44 (64.70%)	47 (79.70%)	
Palate	10 (7.90%)	10 (14.70%)	0 (0.00%)	
Root of the tongue	22 (17.30%)	13 (19.10%)	9 (15.30%)	
Other oropharynx	4 (3.10%)	1 (1.50%)	3 (5.10%)	
Histopathological grading				0.049
G1	7 (8.6%)	6 (12.8%)	1 (2.9%)	
G2	55 (67.9%)	34 (72.3%)	21 (61.8%)	
G3	18 (23.5%)	7 (14.9%)	12 (35.3%)	
Tumor depth (T)				0.743
T1	13 (10.2%)	8 (11.8%)	5 (8.5%)	
T2	42 (33.1%)	24 (35.3%)	18 (30.5%)	
T3	44 (34.6%)	22 (32.4%)	22 (37.3%)	
T4	27 (21.3%)	13 (19.1%)	14 (23.7%)	
Tx	1 (0.8%)	1 (1.5%)	0 (0.0%)	
Lymph node metastasis (N)				0.029
N0	26 (20.50%)	20 (29.40%)	6 (10.20%)	
N1	26 (20.50%)	16 (23.50%)	10 (16.90%)	
N2	57 (44.90%)	24 (35.30%)	33 (55.90%)	
N3	17 (13.40%)	8 (11.80%)	9 (15.30%)	
Nx	1 (0.80%)	0 (0.00%)	1 (1.70%)	
Clinical manifestation				
Pain	73 (57.50%)	39 (57.40%)	34 (57.60%)	0.882
Neck mass	46 (36.20%)	17 (25.00%)	29 (49.20%)	0.008
Cough	6 (4.70%)	5 (7.40%)	1 (1.70%)	0.215
Hoarseness	11 (8.70%)	7 (10.30%)	4 (6.80%)	0.543
Dysphagia	34 (26.80%)	19 (27.90%)	15 (25.40%)	0.906
Dyspnea	2 (1.60%)	2 (2.90%)	0 (0.00%)	0.499
Hemoptysis	6 (4.70%)	4 (5.90%)	2 (3.40%)	0.685
Weight loss	8 (6.30%)	5 (7.40%)	3 (5.10%)	0.724
Hearing impairment	2 (1.60%)	0 (0.00%)	2 (3.40%)	0.214
Duration of medical history (months)	5.55 ± 6.85	5.92 ± 60	5.13 ± 7.26	0.403
Treatment duration (days)	45.88 ± 9.76	43.64 ± 9.00	48.51 ± 10.04	0.005
General treatment regimen				0.003
RT	31 (24.4%)	24 (35.3%)	7 (11.9%)	
CRT	96 (75.6%)	44 (64.7%)	52 (88.1%)	
Detailed treatment regimen				0.006
RT	25 (19.70%)	18 (26.50%)	7 (11.90%)	
CRT	51 (40.20%)	18 (26.50%)	33 (55.90%)	
pRT	6 (4.70%)	6 (8.80%)	0 (0.00%)	
IndCT RT	14 (11.00%)	8 (11.80%)	6 (10.20%)	
IndCT CRT	30 (23.60%)	17 (25.00%)	13 (22.00%)	
Lymph node invasion according to treatment				
RT				0.053
N0–1	23 (74.19%)	20 (87.00%)	3 (13.00%)	
N2–3	8 (25.81%)	4 (50.00%)	4 (50.00%)	
CRT				0.268
N0–1	29 (30.20%)	16 (55.20%)	13 (44.80%)	
N2–3	67 (69.80%)	28 (41.80%)	39 (58.20%)	

Values are presented as means and standard deviations. BMI, body mass index; NRS 2002, Nutritional Risk Score 2002; CRP, C-reactive protein; PNI, prognostic nutritional index, RT, radiotherapy; CRT, chemoradiotherapy; pRT, postoperative radiotherapy; indCT, induction chemoradiotherapy.

**Table 2 cancers-14-02335-t002:** Correlations between regional cancer stage (lymph node metastasis) and treatment type (radiotherapy/chemoradiotherapy).

Feature	RT	CRT	*p*	*p* H–B
All				
N0–1	23 (44.20%)	29 (55.80%)	<0.0001	0.00012
N2–3	8 (10.70%)	67 (89.30%)		
HPV(-)				
N0–1	20 (55.60%)	16 (44.40%)	0.0003	0.0006
N2–3	4 (12.50%)	28 (87.50%)		
HPV(+)				
N0–1	3 (18.75%)	13 (81.25%)	0.3750	0.3750
N2–3	4 (9.30%)	39 (90.70%)		

RT, radiotherapy; CRT, chemoradiotherapy. Fisher’s test. H–B, Holm–Bonferroni correction.

**Table 3 cancers-14-02335-t003:** Weight, BMI, and laboratory results: differences before (0) and after treatment (1) between the HPV(-) and HPV(+) groups.

	HPV(-)	HPV(+)	*p*	H–B
Weight 0–Weight 1 (kg)	4.32 ± 3.71	5.71 ± 4.23	0.099	0.891
Weight 0–Weight 1 (%)	5.65 ± 4.85	7.19 ± 5.19	0.144	1.000
BMI 0–BMI 1	1.54 ± 1.34	1.98 ± 1.40	0.134	1.000
CRP 0–CRP 1	−24.00 ± 44.48	−18.77 ± 23.41	0.431	1.000
Albumin 0–Albumin 1 (g/L)	4.50 ± 4.35	5.02 ± 4.96	0.553	1.000
Prealbumin 0–Prealbumin 1 (g/L)	0.07 ± 0.10	0.08 ± 0.10	0.563	1.000
Hemoglobin 0–Hemoglobin 1 (g/dL)	1.75 ± 1.46	1.85 ± 1.58	0.717	0.717
TLC 0–TLC 1 (/mm^3^)	1.20 ± 0.72	1.37 ± 0.69	0.174	1.000
PNI 0–PNI 1	4.51 ± 4.35	5.92 ± 4.96	0.552	1.000

Values are presented as means and standard deviations. 0–1, difference between values before and after treatment. BMI, body mass index; CRP, C-reactive protein; TLC, total lymphocyte count; PNI, prognostic nutritional index. H–B, Holm–Bonferroni correction for multiple testing.

**Table 4 cancers-14-02335-t004:** BMI, NRS 2002, and laboratory results: differences before and after treatment in the HPV(-) and HPV(+) groups separately.

Feature	All	HPV(-)	HPV(+)
Weight 0 (kg)	75.65 ± 15.24 (44–111)	72.38 ± 15.24 (44–104.80)	79.41 ± 14.64 (46–111)
Weight 1(kg)	71.88 ± 14.17 (42–110)	70.56 ± 15.24 (42–94.70)	72.96 ± 14.81 (45–110)
*p*	<0.0001	<0.0001	<0.0001
*p* H–B	<0.0001	<0.0001	<0.0001
BMI 0 (kg/m^2^)	26.84 ± 4.65 (17.21–40.44)	25.75 ± 4.70 (17.21–37.58)	28.10 ± 4.30 (18.20–40.44)
BMI 1 (kg/m^2^)	25.26 ± 4.16 (17.26–31.81)	24.69 ± 4.04 (17.26–33.96)	25.72 ± 4.24 (17.80–37.81)
*p*	<0.0001	<0.0001	<0.0001
*p* H–B	<0.0001	<0.0001	<0.0001
NRS 2002 0	0.35 ± 0.77 (0–4)	0.41 ± 0.87 (0–4)	0.28 ± 0.64 (0–4)
NRS 2002 1	3.20 ± 1.11 (1–5)	2.98 ± 1.09 (1–5)	3.39 ± 1.10 (1–5)
*p*	<0.0001	<0.0001	<0.0001
*p* H–B	<0.0001	<0.0001	<0.0001
CRP 0 (g/L)	5.15 ± 7.07 (0.16–36.90)	6.79 ± 8.39 (0.16–36.90)	3.40 ± 4.80 (0.17–27.20)
CRP 1 (g/L)	26.60 ± 35.93 (0.17–197.00)	30.96 ± 44.43 (0.20–197.00)	22.00 ± 23.50 (0.17–87.20)
*p*	<0.0001	<0.0001	<0.0001
*p* H–B	<0.0001	<0.0001	<0.0001
Albumin 0 (g/L)	41.67 ± 3.68 (33.00–50.00)	41.58 ± 3.65 (33.00–49.00)	41.77 ± 3.74 (32.00–50.00)
Albumin 1 (g/L)	36.95 ± 3.92 (30.00–49.00)	37.05 ± 4.11 (30.00–47.00)	36.84 ± 3.75 (30.00–49.00)
*p*	<0.0001	<0.0001	<0.0001
*p* H–B	<0.0001	<0.0001	<0.0001
Prealbumin 0 (g/L)	0.28 ± 0.08 (0.13–0.52)	0.26 ± 0.09 (0.13–0.52)	0.29 ± 0.07 (0.18–0.45)
Prealbumin 1 (g/L)	0.20 ± 0.08 (0.08–0.49)	0.19 ± 0.07 (0.08–0.35)	0.21 ± 0.09 (0.09–0.49)
*p*	<0.0001	0.005	0.0004
*p* H–B	<0.0001	0.005	0.0008
Hemoglobin 0 (g/dL)	13.97 ± 1.50 (10.60–17.40)	10.60 ± 1.51 (12.85–15.35)	13.88 ± 1.51 (10.80–16.90)
Hemoglobin 1 (g/dL)	12.70 ± 1.50 (9.40–16.30)	12.29 ± 1.46 (9.40–16.30)	12.03 ± 1.54 (9.50–15.80)
*p*	<0.0001	<0.0001	<0.0001
*p* H–B	<0.0001	<0.0001	<0.0001
TLC 0 (/mm^3^)	1.90 ± 0.72 (0.57–4.84)	1.91 ± 0.71 (0.65–3.94)	1.89 ± 0.73 (0.57–4.54)
TLC 1 (/mm^3^)	0.62 ± 0.37 (0.11–2.84)	0.71 ± 0.44 (0.11–2.84)	0.52 ± 0.24 (0.17–1.34)
*p*	<0.0001	<0.0001	<0.0001
*p* H–B	<0.0001	<0.0001	<0.0001
PNI 0	41.68 ± 3.68 (32.00–50.00)	41.59 ± 3.65 (33.01–49.01)	41.78 ± 3.74 (32.00–50.00)
PNI 1	36.95 ± 3.92 (30.00–49.00)	37.05 ± 4.11 (30.00–47.00)	36.85 ± 3.75 (30.00–49.00)
*p*	<0.0001	<0.0001	<0.0001
*p* H–B	<0.0001	<0.0001	<0.0001

*p*, *p*-value (Wilcoxon test). Values are presented as means and standard deviations. BMI, body mass index; NRS 2002, Nutritional Risk Score 2002; CRP, C-reactive protein; TLC, total lymphocyte count; PNI, prognostic nutritional index; 0, before treatment; 1, after treatment. H–B, Holm–Bonferroni correction.

**Table 5 cancers-14-02335-t005:** Weight, BMI, and laboratory results before and after treatment in the HPV- and HPV+ groups undergoing RT and CRT separately (comparison between HPV groups).

Feature	CRT		RT	
	HPV(-)	HPV(+)	HPV(-)	HPV(+)
Weight 0 (kg)	72.60 ± 14.13	78.58 ± 14.92	71.99 ± 17.04	85.53 ± 11.43
Weight 1(kg)	68.96 ± 13.24	72.10 ± 14.71	75.08 ± 13.44	86.73 ± 9.72
*p*	<0.0001	<0.0001	0.004	0.248
*p* H–B	0.0004	<0.0001	0.062	0.993
BMI 0 (kg/m^2^)	25.11 ± 4.00	27.78 ± 4.31	26.92 ± 5.69	30.52 ± 3.58
BMI 1 (kg/m^2^)	23.56 ± 3.21	25.40 ± 4.10	27.91 ± 4.53	30.91 ± 3.47
*p*	<0.0001	<0.0001	0.004	0.248
*p* H–B	0.0004	<0.0001	0.066	1.000
NRS 2002 0	0.36 ± 0.84	0.20 ± 0.40	0.50 ± 0.93	0.86 ± 1.46
NRS 2002 1	3.06 ± 1.15	3.40 ± 1.12	2.73 ± 0.90	3.33 ± 0.58
*p*	<0.0001	<0.0001	0.004	0.480
*p* H–B	<0.0001	<0.0001	0.058	1.000
CRP 0 (g/L)	7.18 ± 9.76	2.94 ± 3.77	6.02 ± 4.84	6.81 ± 9.24
CRP 1 (g/L)	27.13 ± 35.78	20.66 ± 22.49	38.63 ± 58.38	31.59 ± 30.05
*p*	0.0014	<0.0001	0.014	1.000
*p* H–B	0.025	<0.0001	0.167	1.000
Albumin 0 (g/L)	42.05 ± 3.91	41.92 ± 3.93	40.67 ± 2.96	40.71 ± 1.80
Albumin 1 (g/L)	37.23 ± 4.16	37.02 ± 3.77	36.71 ± 4.10	35.57 ± 3.60
*p*	<0.0001	<0.0001	0.004	0.131
*p* H–B	0.0004	<0.0001	0.062	1.000
Prealbumin 0 (g/L)	0.277 ± 0.079	0.301 ± 0.070	0.230 ± 0.098	0.213 ± 0.029
Prealbumin 1 (g/L)	0.213 ± 0.072	0.217 ± 0.081	0.150 ± 0.048	0.200 ± 0.120
*p*	0.054	0.0003	0.061	1.000
*p* H–B	0.543	0.006	0.552	1.000
Hemoglobin 0 (g/dL)	14.08 ± 1.50	13.85 ± 1.56	14.00 ± 1.54	14.09 ± 1.10
Hemoglobin 1 (g/dL)	11.98 ± 1.40	11.82 ± 1.41	12.87 ± 1.41	13.64 ± 1.65
*p*	<0.0001	<0.0001	0.0002	0.221
*p* H–B	<0.0001	<0.0001	0.003	1.000
TLC 0 (/mm^3^)	1.91 ± 0.78	1.89 ± 0.75	1.91 ± 0.56	1.85 ± 0.57
TLC 1 (/mm^3^)	0.69 ± 0.47	0.53 ± 0.25	0.74 ± 0.38	0.40 ± 0.16
*p*	<0.0001	<0.0001	<0.0001	0.023
*p* H–B	<0.0001	<0.0001	0.0002	0.257
PNI 0	42.06 ± 3.91	41.93 ± 3.93	40.68 ± 2.96	40.72 ± 1.80
PNI 1	37.23 ± 4.16	37.02 ± 3.77	36.72 ± 4.10	35.57 ± 3.60
*p*	<0.0001	<0.0001	0.004	0.131
*p* H–B	0.0003	<0.0001	0.058	0.914

**Table 6 cancers-14-02335-t006:** Overall survival (OS) in HPV-/HPV+ patients: univariate and multivariate analysis.

Variable		OS HPV-					OS HPV+			
		Univariate Analysis		Multivariate Analysis			Univariate Analysis		Multivariate Analysis	
		HR (95%CI)	*p*-Value	HR (95%CI)	*p*-Value		HR (95%CI)	*p*-value	HR (95%CI)	*p*-Value
CRP > 3.50 vs. < 3.50	62	2.08 (0.90–4.82)	0.088	3.45 1.37–8.68	** 0.008 **	58	2.23 (0.70–7.13)	0.176	2.92 0.90–9.46	0.075
Albumin (g/L) > 40.0 vs. < 40.0	62	1.14 (0.50–2.62)	0.755			57	3.26 (0.71–14.89)	0.127	4.21 0.89–19.86	0.070
Prealbumin (g/L) > 0.28 vs. < 0.28	46	0.44 (0.12–1.58)	0.208			44	2.37 (0.59–9.54)	0.223		
Hb 0 (g/dL) > 13.5 vs. < 13.5	68	0.69 (0.32–1.49)	0.345			59	1.31 (0.39–4.37)	0.659		
LC 0 (/mm^3^) > 1.28 vs. < 1.28	68	0.63 (0.26–1.50)	0.299			59	0.63 (0.17–2.35)	0.496		
PNI > 39.01 vs. < 39.01	62	1.04 (0.42–2.53)	0.935			57	1.56 (0.34–7.16)	0.569		
Age > 60 vs. ≤ 60	68	0.788 (0.36–1.71)	0.546			59	0.85 (0.27–2.63)	0.772		
Gender F vs. M		0.49 (0.18–1.31)	0.157	0.12 0.02–0.58	** 0.009 **	59	0.58 (0.16–2.15)	0.414		
General location tonsils vs. others	68	1.57 (0.68–3.63)	0.289							
Tumor depth (T) T3–4 vs. T1–2	67	0.95 (0.44–2.06)	0.899			59	1.79 (0.48–6.62)	0.385		
Lymph node metastasis N2–3 vs. N0–1	68	1.80 (0.83–3.93)	0.139			58	1.15 (0.31–4.26)	0.832		
Radiotherapy vs. chemoradiotherapy	68	0.93 (0.42–2.06)	0.858			59	0.60 (0.08–4.64)	0.623		
BMI 0 > 24.53 vs. < 24.53	68	0.39 (0.17–0.87)	** 0.022 **			59	0.57 (0.15–2.12)	0.400		
NRS 2002 > 1 vs. ≤ 1		4.03 (1.56–10.37)	** 0.004 **	3.83 1.22–12.01	** 0.021 **	58	0.00	1.00		
Smoking Yes vs. no	68	1.18 (0.49–2.84)	0.706			59	0.63 (0.17–2.32)	0.483		
Alcohol abuse Yes vs. no	68	4.17 (1.20–14.47)	0.025	6.39 1.40–29.14	** 0.017 **					

0, before treatment; Hb, hemoglobin level; TLC, lymphocyte count.

**Table 7 cancers-14-02335-t007:** Disease-free survival (DFS) in HPV-/HPV+ patients: univariate and multivariate analysis.

Variable		DFS HPV-					DFS HPV+			
		Univariate Analysis		Multivariate Analysis			Univariate Analysis		Multivariate Analysis	
		HR (95%CI)	*p*-Value	HR (95%CI)	*p*-Value		HR (95%CI)	*p*-Value	HR (95%CI)	*p*-Value
CRP >3.50 vs. <3.50	62	2.78 1.13–6.82	** 0.026 **	2.90 1.08–7.76	** 0.034 **	58	4.95 1.18–20.76	0.029		
Albumin (g/L) >40.0 vs. <40.0	62	0.902 0.39–2.09	0.810			57	0.96 0.23–4.05	0.957		
Prealbumin (g/L) >0.28 vs. <0.28	46	0.50 0.14–1.79	0.288			44	1.01 0.25–4.02	0.993		
Hb 0 (g/dL) >13.5 vs. <13.5	**68**	**0.38** **(0.17–0.87)**	** 0.021 **	0.34 0.13–0.90	** 0.029 **	59	1.15 (0.27–4.82)	0.848		
TLC 0 (/mm^3^) >1.28 vs. <1.28	68	0.61 (0.25–1.47)	0.271			**59**	**0.19** **(0.05–0.77)**	** 0.020 **	**0.20** **(0.05–0.82)**	** 0.025 **
PNI > 39.01 vs. < 39.01	62	0.55 0.23–1.28	0.163			57	0.45 0.11–1.89	0.276		
Age > 60 vs. ≤ 60	68	0.75 0.34–1.65	0.474			59	0.88 0.22–3.53	0.859		
Gender F vs. M	68	0.99 0.41–2.39	0.984			**59**	**0.95** **0.22–3.96**	**0.941**		
General location tonsils vs. others	**68**	**1.35** **0.58–3.15**	**0.482**			59		1.00		
Tumor depth (T) T3–4 vs. T1–2	67	1.20 0.54–2.64	0.655			**59**	**2.02** **0.41–10.00**	**0.390**		
Lymph node metastasisN2–3 vs. N0–1	**68**	**3.55** **(1.51–8.34)**	** 0.004 **	**4.26** **(1.59–11.43)**	** 0.004 **	58	2.86 (0.35–23.24)	0.326		
Radiotherapy vs. chemoradiotherapy	68	0.71 (0.31–1.64)	0.422				0.00	1.00		
BMI 0 >24.53 vs. <24.53	68	0.41 0.18–0.91	** 0.028 **			**59**	**0.377** **0.09–1.58**	**0.182**		
NRS 2002 >1 vs. ≤1	**68**	**4.46** **1.74–11.45**	** 0.002 **	**5.89** **1.79–19.37**	** 0.003 **	58	0.00	1.00		
Smoking Yes vs. no	68	0.88 0.39–1.99	0.760			59	1.20 0.29–5.03	0.801		
Alcohol abuse Yes vs. no	68	2.92 0.68–12.59	0.150	8.01 1.41–45.39	** 0.019 **	59		1.00		

0, before treatment; Hb, hemoglobin level; TLC, lymphocyte count.

## Data Availability

The data presented in this study are available in this article and Appendix A available online.

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
