# Peer review of "Analysis of Selected Nutritional Parameters in Patients with HPV-Related and Non-HPV-Related Oropharyngeal Cancer before and after Radiotherapy Alone or Combined with Chemotherapy"

_cancers, 2022, doi:10.3390/cancers14092335_

Round 1

Reviewer 1 Report

Why were parameters non significant at univariate analysis included in multivariate analysis and how was it possible that some of them became significant?

Author Response

Dear Reviewer,

This is the revised invited original article „Analysis of selected nutritional parameters in patients with HPV-related and HPV- not related oropharyngeal cancer before and after radiotherapy alone or combined with chemotherapy” (cancers-1666344) for the special issue "Diagnostic, Prognostic, Predictive Biomarkers and New Targets for Treatment in Head and Neck Cancers" of the “Cancers” journal.

Thank you for your questions and comments. We have fully addressed all the comments and our responses appear below. Our revised work includes corrections according to reviewers’ comments in the text. Our revisions, made according to reviewers’ comments, are marked using the „Track Changes” function in the main manuscript and Supplementary materials.

We take this opportunity to express our gratitude to the reviewers for their constructive and useful remarks. Their comments allowed us to identify areas in my manuscript that needed modification.

We also thank you for allowing us to resubmit a revised copy of the manuscript.

We hope that the revised manuscript is now suitable for publication in Cancers.

The manuscript is original and it has not been published or accepted for publication, either in whole or in part, in any form. No part of the manuscript is currently under consideration for publication elsewhere.

Yours sincerely,

Beata Jabłońska, MD, PhD.

Responses to Reviewers’ comments

Reviewer 1

Comment:

Why were parameters non significant at univariate analysis included in multivariate analysis and how was it possible that some of them became significant?

Answer:

In our analysis, after performing the univariate analysis, we included all parameters in the multivariate analysis and, using backward stepwise regression, we reduced the variables under consideration to those most influencing survival. There are different approaches to multivariate analysis. Either all variables and back reduction, or selecting only those variables that are significant in univariate, or selecting variables that have p-value <0.2 or 0.25 in univariate and include them in multivariate analysis. (Hosmer DW, Lemeshow S. Applied survival analysis).

In the other words, the univariate Cox model assesses the effect of individual variables on survival separately. Multivariate analysis assesses the impact on survival of all variables taken together. In our paper, the Cox model adjusts all variables simultaneously, and therefore may non-significant variables in the univariate analysis become statistically significant in the multivariate analysis, and vice versa. After reduction of variables (backward stepwise regression) to variables with the strongest impact on survival, the final model was obtained. This model includes variables that carry independent information. Below we are presenting an example of changes in the assessment of the separate impact of individual variables on survival and the assessment of the combined impact of these variables in Table S1 (currently Table S5 in the revised version). Overall survival (OS) in patients: univariate and multivariate analysis for pre-treatment parameters. This table shows: multivariate analysis preliminary model  including all obtained results and multivariate analysis reduced model presenting in the paper.

Table S5. Overall survival (OS) in patients: univariate and multivariate analysis for pre-treatment parameters

Variable

Univariate analysis

Multivariate analysis preliminary model  n=116

Multivariate analysis
reduced model

n

HR
(95%CI)

p-value

HR
(95%CI)

p-value

HR
(95%CI)

p-value

CRP
>3.50 vs. 3.50

120

2.50
(1.28-4.88)

0.007

2.73
(1.22-6.15)

0.015

2.22
1.12-4.42

0.023

Albumin [g/l]
>40.0 vs. <40.0

119

1.41
(0.70-2.83)

0.337

1.71
(0.45-6.55)

0.432

Hb 0 [g/dl] 
>13.5 vs. <13.5

127

0.82
(0.43-1.55)

0.544

0.81
(0.35-1.86)

0.615

TLC 0 [/mm3] 
>1.28 vs. <1.28

127

0.62
(0.30-1.28)

0.196

0.71
(0.31-1.65)

0.428

PNI >39.01 vs. <39.01

119

1.08
(0.50-2.30)

0.845

0.71
(0.18-2.81)

0.626

Age  >60 vs. ≤60

127

0.86
(0.46-1.63)

0.653

0.91
(0.40-2.05)

0.816

Gender F vs. M

127

0.49
(0.22-1.07)

0.073

0.26
(0.09-0.71)

0.009

0.32
0.13-0.82

0.017

General location
tonsil vs. others

127

1.60
(0.73-3.49)

0.239

3.07
(1.04-9.08)

0.043

3.20
1.19-8.60

0.021

Tumor depth (T)
T3-4 vs. T1-2

126

1.02
(0.53-1.94)

0. 962

0.70
(0.32-1.54)

0.381

Lymph node metastasis
N 2-3 vs. N 0-1

126

1.09
(0.57-2.09)

0.791

1.09
(0.46-2.57)

0.851

Radiotherapy vs.
Radiochemotherapy

127

1.28
(0.63-2.58)

0.493

1.24
(0.45-3.39)

0.682

BMI 0
>24.53 vs. <24.53

127

0.37
(0.19-0.70)

0.003

0.35
(0.13-0.95)

0.039

0.32
0.14-0.72

0.006

NRS 2002
>1 vs. ≤1

126

3.99
(1.66-9.62)

0.002

1.29
0.37-4.53)

0.688

Smoking
Yes vs. No

127

1.40
(0.74-2.67)

0.306

0.53
(0.20-1.39)

0.198

0.48
0.22-1.07

0.074

Alcohol abuse
Yes vs. No

127

6.38
(1.89-21.51)

0.003

3.21
(0.76-13.53)

0.112

3.86
1.00-14.86

0.050

HPV+ vs. HPV-

127

0.31
(0.16-0.63)

0.001

0.44
(0.17-1.11)

0.081

0.36
0.16-0.80

0.012

Table S5. Overall survival (OS) in patients: univariate and multivariate analysis for pre-treatment parameters

Variable

Univariate analysis

Multivariate analysis

n

HR
(95%CI)

p-value

HR
(95%CI)

p-value

CRP
>3.50 vs. 3.50

120

2.50
(1.28-4.88)

0.007

2.22
1.12-4.42

0.023

Albumin [g/l]
>40.0 vs. <40.0

119

1.41
(0.70-2.83)

0.337

Hb 0 [g/dl] 
>13.5 vs. <13.5

127

0.82
(0.43-1.55)

0.544

TLC 0 [/mm3] 
>1.28 vs. <1.28

127

0.62
(0.30-1.28)

0.196

PNI >39.01 vs. <39.01

119

1.08
(0.50-2.30)

0.845

Age  >60 vs. ≤60

127

0.86
(0.46-1.63)

0.653

Gender F vs. M

127

0.49
(0.22-1.07)

0.073

0.32
0.13-0.82

0.017

General location
tonsil vs. others

127

1.60
(0.73-3.49)

0.239

3.20
1.19-8.60

0.021

Tumor depth (T)
T3-4 vs. T1-2

126

1.02
(0.53-1.94)

0. 962

Lymph node metastasis
N 2-3 vs. N 0-1

126

1.09
(0.57-2.09)

0.791

Radiotherapy vs.
Radiochemotherapy

127

1.28
(0.63-2.58)

0.493

BMI 0
>24.53 vs. <24.53

127

0.37
(0.19-0.70)

0.003

0.32
0.14-0.72

0.006

NRS 2002
>1 vs. ≤1

126

3.99
(1.66-9.62)

0.002

Smoking
Yes vs. No

127

1.40
(0.74-2.67)

0.306

0.48
0.22-1.07

0.074

Alcohol abuse
Yes vs. No

127

6.38
(1.89-21.51)

0.003

3.86
1.00-14.86

0.050

HPV+ vs. HPV-

127

0.31
(0.16-0.63)

0.001

0.36
0.16-0.80

0.012

0, before treatment; Hb, haemoglobin level;  TLC, lymphocyte count .

Reviewer 2 Report

The authors added lots of statistical analyses and correction of p values according to the reviewer's comments. But some tables seem not to be essential (especially Table 5, 6, 8, and 9), leading to a detour to conclusions. The multivariate analyses for OS demonstrated that only nutritional parameters of BMI and NRS were related to OS of HPV(-)OPC, but not in HPV(+)OPC (Table 10). I respect authors' lots of analyses, but their findings have less impact on strategy of CRT for OPC.

Author Response

Dear Reviewer,

This is the revised invited original article „Analysis of selected nutritional parameters in patients with HPV-related and HPV- not related oropharyngeal cancer before and after radiotherapy alone or combined with chemotherapy” (cancers-1666344) for the special issue "Diagnostic, Prognostic, Predictive Biomarkers and New Targets for Treatment in Head and Neck Cancers" of the “Cancers” journal.

Thank you for your questions and comments. We have fully addressed all the comments and our responses appear below. Our revised work includes corrections according to reviewers’ comments in the text. Our revisions, made according to reviewers’ comments, are marked using the „Track Changes” function in the main manuscript and Supplementary materials.

We take this opportunity to express our gratitude to the reviewers for their constructive and useful remarks. Their comments allowed us to identify areas in my manuscript that needed modification.

We also thank you for allowing us to resubmit a revised copy of the manuscript.

We hope that the revised manuscript is now suitable for publication in Cancers.

The manuscript is original and it has not been published or accepted for publication, either in whole or in part, in any form. No part of the manuscript is currently under consideration for publication elsewhere.

Yours sincerely,

Beata Jabłońska, MD, PhD.

Responses to Reviewers’ comments

Reviewer 2

Comment:

The authors added lots of statistical analyses and correction of p values according to the reviewer's comments. But some tables seem not to be essential (especially Table 5, 6, 8, and 9), leading to a detour to conclusions.

Answer:

According to this Reviewer’s suggestion, but taking into account the two other Reviewers’ comments (who appreciated the added statistical analyses and results and did not suggest to remove above mentioned tables) as well as taking into account below mentioned points, we have removed Table 5, 6, 8, and 9 from the main manuscript and inserted them in the Supplementary Materials as Table S1, S2, S3, S4. The Table S1 and S2 have been re-numbered as Table S5 and S6.

Table 5 and 9 had been presented already in the primary version of our paper. They present important results consistent with the main topic of the work.

In Table 5, laboratory results in patients before and after treatment were presented. In our opinion, comparison of pre-treatment and post-treatment laboratory results is important, and the results are also discussed in the text.

In Table 9, Comparison of selected clinicopathological factors between groups with low and high prognostic nutritional index (PNI) is presented. PNI is an important nutritional parameter. Therefore, in our opinion, the table comparing low- and high-PNI groups is also essential. The results are commented also in the text.

Tables 6 and 8 have been added according to the reviewer's comments in order to exclude the impact of comcomitant chemotherapy on final results. Taking into account the Reviewers’ suggestions, additional analyses for RT and CRT groups have been performed to exclude the impact of comcomitant chemotherapy on final results (Tables 6-8).

Table 6 presents weight, BMI and laboratory results: differences before and after treatment in HPV- and HPV+ groups undergoing RT and CRT separately (comparison of values before and after RT/CRT within individual HPV- and HPV+ subgroups + separately).

Table 8 presents weight, BMI and laboratory results before and after treatment in HPV- and HPV+ groups undergoing RT and CRT separately (comparison of difference values between two subgroups according to HPV status).

In our opinion, all presented comparisons of HPV- and HPV+ OPC patients (including all patients and separate groups treated by RT or CRT) are necessary, because they show that HPV+ status associated with more frequent CRT (but not HPV status alone) predicts the greater deterioration of NS in OPC patients.

Using all presented analyses, we have shown that a higher decrease of most nutritional parameters was observed in HPV+ compared to HPV- group that was associated with the more frequent CRT in this group. RT alone, as the only treatment, was more common in HPV- patients mostly due to less advanced stages of the disease. But, in comparison of NS in HPV- and HPV+ OPC patients following RT and CRT separately, this difference was not found. CRT adversely affected NS of patients regardless of HPV status. RT caused a significant decrease of haemoglobin concentration and TLC in HPV- patients. There was no significant difference regarding other nutritional parameters after RT in both groups. To state above mentioned conclusions, all statistical analyses were needed. According to the one Reviewer and Editor's suggestions we have removed from the main manuscript and trasferred the indicated tables to Supplementary Materials.

Comment:

The multivariate analyses for OS demonstrated that only nutritional parameters of BMI and NRS were related to OS of HPV(-)OPC, but not in HPV(+)OPC (Table 10).

Answer:

The aim of our study was not only to analyze the impact of nutritional parameters on patients’ survival, but mainly comparison of NS between HPV- and HPV+ OPC patients and analysis of the impact of RT/CRT on NS in HPV- and HPV+ OPC patients. The multivariate analyses for OS demonstrating that only nutritional parameters of BMI and NRS were related to OS of HPV(-)OPC, but not in HPV(+)OPC (Table 10, currently Table 6) are not the only essential result in our study. OS and DFS were significantly better in patients with NRS 2002<2 compared to NRS 2002≥2 for HPV- patients and not significantly for the HPV+ group.

Regarding results obtained in the multivariate Cox analysis, in HPV+ patients, higher TLC was a significant independent good prognostic factor for DFS (Table 11, currently Table 7).

Moreover, as it was above mentioned, we presented comprehensive comparisons of pre-treatment and post-treatment NS between HPV- and HPV+ OPC patients as well as comparisons of NS deterioration following RT/CRT between HPV- and HPV+ OPC patients. CRT caused significant weight loss and decrease of BMI, albumin, TLC, haemoglobin concentration and increase of NRS 2002 in HPV- and HPV+ patients. A significant decrease of prealbumin level after CRT was noted only in HPV+ patients. RT caused a significant decrease of haemoglobin concentration and TLC in HPV- patients. There was no significant difference regarding other parameters after RT in both groups. RT did not have negative impact on BMI, weight, NRS, CRP, Alb, prealbumin and PNI.

Comparison of differences of weight, BMI, NRS, PNI and laboratory results before and after CRT and RT between HPV- and HPV+ patients did not show any significant differences between groups according to HPV status.

 All presented in the study findings (reporting or do not reporting differences between compared groups) are clinically relevant and may be used in the clinical practice.

All essential results of our study are described in the section “Materials and methods”. To emphasize above mentioned points, we have added some more important results in the abstract. Currently, the most important findings of our study are presented in the abstract as follows:

Results: In both groups, a significant decrease in all analyzed nutritional parameters was noted after RT/CRT (p<0.01). CRT caused significant weight loss and decrease of BMI, albumin, total lymphocyte count (TLC), haemoglobin concentration and increase of Nutritional Risk Score (NRS) 2002 in HPV- and HPV+ patients. A significant decrease of prealbumin level after CRT was noted only in HPV+ patients. RT caused a significant decrease of haemoglobin concentration and TLC in HPV- patients. There was no significant difference regarding other nutritional parameters after RT in both groups. RT did not have negative impact on body mass index (BMI), weight, NRS, CRP, Alb, Prealb and PNI. Overall survival (OS) and disease-free survival (DFS) were significantly better in patients with a higher BMI in the HPV- group (OS, p=0.011) and (DFS, p=0.028); DFS was significantly better in patients with C-reactive protein (CRP) <3.5 g/dl in HPV- (p=0.021) and HPV+ (p=0.018) groups, and total lymphocyte count (TLC)>1.28 /mm3 in the HPV+ group (p=0.014). Higher NRS 2002 was an independent adverse prognostic factor for OS and DFS in HPV-, but not in the HPV+ group. Kaplan Meier analysis showed  that both OS and DFS were significantly better in HPV- patients with lower NRS 2002 score. However, this relationship was not observed in the HPV+ group.

Comment:

I respect authors' lots of analyses, but their findings have less impact on strategy of CRT for OPC.

Answer:

Although our  findings may have less impact on strategy of CRT for OPC, but they have strong impact on nutritional support during CRT for OPC and it is the main goal of our work. Our study showed that both HPV- and HPV+ OPC patients can develop malnutrition during RT/CRT. Therefore, regardless HPV status, nutritional support is required in OPC patients during RT/CRT. It is known that optimal strategy regarding nutritional support in OPC patients is as important as strategy of CRT, because OPC patients are malnourished due to OPC and oncological treatment. Our results are an important voice in the worldwide discussion regarding the impact of the HPV status on NS in OPC patients. There are only few and contradictory reports on this topic in the literature. Our study is one of the first publications regarding influence of HPV status on NS in patients with OPC before and after RT/CRT. To our knowledge, it is the first study analyzing impact of nutritional parameters on survival depending on HPV status.

Reviewer 3 Report

The authors have improved the language quality and included more results to support their conclusion. The self-plagiarism has been removed from the previous submission.

Author Response

Dear Reviewer,

This is the revised invited original article „Analysis of selected nutritional parameters in patients with HPV-related and HPV- not related oropharyngeal cancer before and after radiotherapy alone or combined with chemotherapy” (cancers-1666344) for the special issue "Diagnostic, Prognostic, Predictive Biomarkers and New Targets for Treatment in Head and Neck Cancers" of the “Cancers” journal.

Thank you for your questions and comments. We have fully addressed all the comments and our responses appear below. Our revised work includes corrections according to reviewers’ comments in the text. Our revisions, made according to reviewers’ comments, are marked using the „Track Changes” function in the main manuscript and Supplementary materials.

We take this opportunity to express our gratitude to the reviewers for their constructive and useful remarks. Their comments allowed us to identify areas in my manuscript that needed modification.

We also thank you for allowing us to resubmit a revised copy of the manuscript.

We hope that the revised manuscript is now suitable for publication in Cancers.

The manuscript is original and it has not been published or accepted for publication, either in whole or in part, in any form. No part of the manuscript is currently under consideration for publication elsewhere.

Yours sincerely,

Beata Jabłońska, MD, PhD.

Responses to Reviewers’ comments

Reviewer 3

Comment:

The authors have improved the language quality and included more results to support their conclusion. The self-plagiarism has been removed from the previous submission.

Answer:

Thank you for appreciating our work. Thank you very much for your positive opinion.

Round 2

Reviewer 1 Report

Thanks for responding to the issue.

Author Response

Dear Reviewer,

This is the revised invited original article „Analysis of selected nutritional parameters in patients with HPV-related and HPV- not related oropharyngeal cancer before and after radiotherapy alone or combined with chemotherapy” (cancers-1666344) for the special issue "Diagnostic, Prognostic, Predictive Biomarkers and New Targets for Treatment in Head and Neck Cancers" of the “Cancers” journal.

Thank you for your questions and comments. We have fully addressed all the comments and our responses appear below. Our revised work includes corrections according to reviewers’ comments in the text. Our revisions, made according to reviewers’ comments, are marked using the „Track Changes” function in the main manuscript and Supplementary materials.

We take this opportunity to express our gratitude to the reviewers for their constructive and useful remarks. Their comments allowed us to identify areas in my manuscript that needed modification.

We also thank you for allowing us to resubmit a revised copy of the manuscript.

We hope that the revised manuscript is now suitable for publication in Cancers.

The manuscript is original and it has not been published or accepted for publication, either in whole or in part, in any form. No part of the manuscript is currently under consideration for publication elsewhere.

Yours sincerely,

Beata Jabłońska, MD, PhD.

Responses to Reviewers’ comments

Reviewer 1

Comment:

Thanks for responding to the issue.

Answer:

Thank you for appreciating our work. Thank you very much for your positive opinion.

Reviewer 2 Report

The authors precisely addressed the reviewer's comments. The revised version is well refined. Their findings and opinions became clear, leading to readers' understanding. Some minor issues should be revised.

The period the patients were enrolled is not mentioned.

It should be described clearly whether this study is prospective or retrospective in Material and Method section. In Line 561, "Although, all parameters, including HPV testing and stratification of our patients, were collected prospectively, statistical analysis of collected data was retrospective." This is the first showing up about pro- or retro- in the manuscript.

The details about chemo-radiation, and induction chemotherapy are unclear. Detailed chemotherapeutic agents and doses should be shown in the manuscript.

Table 1, <18.5 is repeated.

Line 237, (-5.65±4.85 (-19.35-4.84 vs. (- 7.19±5.19 (-20.56-1.85)); Parentheses are strange.

Line 237 CRT in was similar....   [in] is not necessary.

Table 3, The values should indicate parameter 0 -1, e.g. CRP 0 - CRP 1. It should be described clearly in the legends.

Line 272-,

for TLC 0 & CRP 0 "–" R=0.053 (p=0.563); for TLC 1 & CRP 1 "  " R= -0.116 (p=0.214). The expression should be unified.

Line 338, There was better OS in patients with CRP<1.27 g/dl in all patients (p=0.002) and HPV+ group (p=0.005).

Line 345, There was no statistical difference between OS and DFS according to loss of weight in the both groups (p>0.05).

Line 347, there was no statistically significant difference in OS and DFS, when patientswere divided into two subgroups (NRS 2002 <3 vs. NRS 2002 ≥3) in both HPV- and HPV+ patients (p>0.05).

These results are not found in the figures, leading to readers' confusion.

Line 440-, Authors observed cerebrovascular disease"," peripheral vascular disease, dementia, ulcer disease, "or" liver disease most often in HPV- OPC patients.

Line 451-, loss of appetite"," odynophagia,

Line 581, "The greater association between NS and survival was noted in HPV- patients." Why not observed in HPV+? Maybe, HPV(+) is a stronger factor on survival than nutritional parameters. Reasons should be discussed in the manuscript.

Author Response

Dear Reviewer,

This is the revised invited original article „Analysis of selected nutritional parameters in patients with HPV-related and HPV- not related oropharyngeal cancer before and after radiotherapy alone or combined with chemotherapy” (cancers-1666344) for the special issue "Diagnostic, Prognostic, Predictive Biomarkers and New Targets for Treatment in Head and Neck Cancers" of the “Cancers” journal.

Thank you for your questions and comments. We have fully addressed all the comments and our responses appear below. Our revised work includes corrections according to reviewers’ comments in the text. Our revisions, made according to reviewers’ comments, are marked using the „Track Changes” function in the main manuscript and Supplementary materials.

We take this opportunity to express our gratitude to the reviewers for their constructive and useful remarks. Their comments allowed us to identify areas in my manuscript that needed modification.

We also thank you for allowing us to resubmit a revised copy of the manuscript.

We hope that the revised manuscript is now suitable for publication in Cancers.

The manuscript is original and it has not been published or accepted for publication, either in whole or in part, in any form. No part of the manuscript is currently under consideration for publication elsewhere.

Yours sincerely,

Beata Jabłońska, MD, PhD.

Responses to Reviewers’ comments

Reviewer 2

Comment:

The authors precisely addressed the reviewer's comments. The revised version is well refined. Their findings and opinions became clear, leading to readers' understanding. Some minor issues should be revised.

Answer:

Thank you for appreciating our work. Thank you very much for your positive opinion. Some minor issues have been revised according to the Reviewer’s suggestions. Thay are presented below.

 Comment:

The period the patients were enrolled is not mentioned.

 Answer:

It has been mentioned in the paragraph „Materials and Methods” as follows:

The analysis included 127 patients with OPC who received definitive radical RT/CRT at I Radiation and Clinical Oncology Department of Maria Skłodowska-Curie Research Institute of Oncology, Gliwice Branch, Poland in the period 2012-2016.

Comment:

It should be described clearly whether this study is prospective or retrospective in Material and Method section. In Line 561, "Although, all parameters, including HPV testing and stratification of our patients, were collected prospectively, statistical analysis of collected data was retrospective." This is the first showing up about pro- or retro- in the manuscript.

 Answer:

This study is a retrospective analysis of nutritional parameters of patients from the clinical database that had been collected prospectively in the real time. According to the Reviewer’s suggestion, the sentence "Although, all parameters, including HPV testing and stratification of our patients, were collected prospectively, statistical analysis of collected data was retrospective" has been removed. Only the sentence „The single center observation and retrospective analysis are limitations of this study” has been presented in the manuscript.

Comment:

The details about chemo-radiation, and induction chemotherapy are unclear. Detailed chemotherapeutic agents and doses should be shown in the manuscript.

Answer:

Detailed chemotherapeutic agents and doses should have been shown in the paragraph 2.2. Study design as follows:

Simultaneously, during radiotherapy, Cisplatin was administered at a dose of 100 mg / m2 on irradiation days 1, 22, and 43 or at a dose of 40 mg / m2 administered weekly. In the case of induction chemotherapy, 2-3 cycles were used according to the PF regimen (Cisplatin and 5-Fluoruracil) or the TPF regimen (Docetaxel, Cispaltin and 5-Fluorouracil).

 Comment:

Table 1, <18.5 is repeated.

Answer:

It has been corrected as follows:

< 18.5

> 18.5

 Comment:

Line 237, (-5.65±4.85 (-19.35-4.84 vs. (- 7.19±5.19 (-20.56-1.85)); Parentheses are strange.

Answer:

 It has been corrected as folows:

: -5.65±4.85 (-19.35-4.84) vs. -7.19±5.19 (-20.56-1.85);

Comment:

Line 237 CRT in was similar....   [in] is not necessary.

 Answer:

It has been removed as follows:

Loss of weight (LOW) during CRT was similar in HPV- and HPV+ patients: -5.65±4.85 (-19.35-4.84) vs. -7.19±5.19 (-20.56-1.85); p=0.144, pH-B=1.000 (Table 3).

Comment:

Table 3, The values should indicate parameter 0 -1, e.g. CRP 0 - CRP 1. It should be described clearly in the legends.

Answer:

The values 0-1 of parameters have been presented in Table S3. In Table 3, only the calculated differences between parameters before treatment (0) and after treatment (1) have been presented. According to the Reviewer’s suggestion, the clear description has been added in the legends as follows:

Table 3. Weight, BMI and laboratory results: differences before (01) and after treatment (1): comparison between HPV(-) and HPV(+) groups.

HPV(-)

HPV(+)

p

H-B

Weight 0 – Weight 1[kg]

4.32±3.71

5.71±4.23

0.099

0.891

Weight 0 – Weight 1 [%]

5.65±4.85

7.19±5.19

0.144

1.000

BMI 0 – BMI 1

1.54±1.34

1.98±1.40

0.134

1.000

CRP 0 -CRP 1

-24.00±44.48

-18.77±23.41

0.431

1.000

Albumin 0 – Albumin 1  [g/l]

4.50±4.35

5.02±4.96

0.553

1.000

Prealbumin 1 – Prealbumin 1 [g/l]

0.07±0.10

0.08±0.10

0.563

1.000

Haemoglobin 1 – Hemoglobin 1 [g/dl]

1.75±1.46

1.85±1.58

0.717

0.717

TLC 1 – TLC 1 [/mm3]

1.20±0.72

1.37±0.69

0.174

1.000

PNI 0 – PNI 1

4.51±4.35

5.92±4.96

0.552

1.000

Values are presented as means and standard deviations.

0-1; difference between value before and after treatment.

BMI, body mass index; CRP, C-reactive protein; TLC, total lymphocyte count; PNI, prognostic nutritional index.

H-B Holm-Bonferroni correction for multiple testing

 Comment:

Line 272-,

for TLC 0 & CRP 0 "–" R=0.053 (p=0.563); for TLC 1 & CRP 1 "  " R= -0.116 (p=0.214). The expression should be unified.

Answer:

It has been unified as follows:

The correlations between TLC and CRP before and after treatment were analyzed and excluded: for TLC 0 & CRP 0: R=0.053 (p=0.563); for TLC 1 & CRP 1: R= -0.116 (p=0.214), in Spearman test.

Comment: 

Line 338, There was better OS in patients with CRP<1.27 g/dl in all patients (p=0.002) and HPV+ group (p=0.005).

Answer:

The sentence structure has been changed and  statistical values reported in the text and figures have been unified as follows:

There was better OS in patients with CRP<3.50 g/dl in whole cohort (p=0.008).

Comment:

Line 345, There was no statistical difference between OS and DFS according to loss of weight in the both groups (p>0.05).

Answer:

This information has been more clearly presented based on results shown in Figure S1 that has been added in the current Supplementary Materials. We have added it to the Supplementary Materials, because these difference are not statistically significant.

Also, a reference to this figure has been added in the text as follows:

There was no statistical difference between HPV- and HPV+ patients regarding OS and DFS according to loss of weight (<5% vs >5%) in the both groups (p>0.05) (Figure S1).

Figure S1. Overall and disease-free survival in HPV- and HPV+ patients depending on NRS 2002 (NRS 2002<3 vs. NRS 2002≥3).

Comment:

Line 347, there was no statistically significant difference in OS and DFS, when patientswere divided into two subgroups (NRS 2002 <3 vs. NRS 2002 ≥3) in both HPV- and HPV+ patients (p>0.05).

These results are not found in the figures, leading to readers' confusion.

 Answer:

Figure S2 has been added in order to show these results. We have added it to the Supplementary Materials, because these difference are not statistically significant.

Also, a reference to this figure has been added in the text as follows:

Regarding NRS 2002, there was no statistically significant difference in OS and DFS, when patients were divided into two subgroups (NRS 2002 <3 vs. NRS 2002 ≥3) in both HPV- and HPV+ patients (p>0.05) (Figure S2).

Figure S2. Overall and disease-free survival in HPV- and HPV+ patients depending on NRS 2002 (weight loss<5% vs. weight loss>5%).

Comment:

Line 440-, Authors observed cerebrovascular disease"," peripheral vascular disease, dementia, ulcer disease, "or" liver disease most often in HPV- OPC patients.

Answer:

It has been corrected according to the Reviewer’s suggestion as follows:

Authors observed cerebrovascular disease, peripheral vascular disease, dementia, ulcer disease, or liver disease most often in HPV- OPC patients.

Comment:

Line 451-, loss of appetite"," odynophagia,

It has been corrected according to the Reviewer’s suggestion as follows:

Predominant acute side effect, like xerostomia, mucositis, loss of appetite, odynophagia, dysphagia, nausea and vomiting, usually lead to weight loss and dehydration.

Comment:

Line 581, "The greater association between NS and survival was noted in HPV- patients." Why not observed in HPV+? Maybe, HPV(+) is a stronger factor on survival than nutritional parameters. Reasons should be discussed in the manuscript.

Answer:

We are agree with this Reviewer’s suggestion (even this reason had been presented in the primary version of our manuscript, but according to another Reviewer’s suggestion, it has been removed). According to the current Reviewer’s suggestion, this reason has been re-added as supplemented using the Reviewer’s suggestion in the discussion and conclusions as follows:

Discussion:

It should be added that our study did not show any clear impact of the nutritional status (including PNI) on survival in patients with OPC regardless HPV infection. It may be associated with the other stronger factors, including HPV status, which determine survival in patients with OPC. Maybe, positive HPV status is a stronger factor on survival than nutritional parameters.This hypothesis is supported by a fact that our study showed the greater association between NS and survival in HPV- patients.

Conclusions:

The greater association between NS and survival was noted in HPV- patients. It might be associated with a fact that HPV+ is a stronger factor for survival than nutritional parameters.

This manuscript is a resubmission of an earlier submission. The following is a list of the peer review reports and author responses from that submission.

Round 1

Reviewer 1 Report

This study was designed to compare nutritional status in patients with HPV+ and HPV- OPC, before and after RT or CRT. This manuscript was prepared by the authors who just published an article in Cancers (Basel). 2021 Jun 29;13(13):3256. Comparison of Selected Immune and Hematological Parameters and Their Impact on Survival in Patients with HPV-Related and HPV-Unrelated Oropharyngeal Cancer. Obviously, they used the same sample, and the same study design to analyze their results, as most of the results are totally the same, especially Table 1, 2,  Figure 4. This significantly draws back the novelty of this study.

Some results don’t support their conclusions. Regarding NRS 2002, significantly inferior OS and DFS were noted in HPV- patients (but not in HPV+  patients) in the division (NRS 2002 <2 vs. NRS 2002 ≥2) (Figure 3). Why the authors explained, “This phenomenon was not reported in HPV+ patients, because HPV status was the strongest prognostic factor for survival in this patients’ group (Figure 4).”

Too many errors in the manuscripts, like line 79, between patients with HPV- and HPV- OPC, line 190, HPV- and HPV groups, line 202, LOW…  The authors need to improve the manuscript significantly.

Reviewer 2 Report

This is an interesting study about nutritional parameters in patients with HPV-related and HPV- not related oropharyngeal cancer before and after radiotherapy.

The paper is well written. However, some issues remain.

I think that oropharyngeal carcinoma should be a better definition than oropharyngeal cancer.

In materials and methods section, the authors must report which laboratory blood tests were analyzed.

Post-treatment evaluations were performed at different follow-up time. This represents a bias and must be discussed. Moreover, patients whose post-operative assessment was performed too soon or too late compared to end of the treatment, must be excluded from the analyses.

General tumor location should be removed from table 1. The detailed tumor location is sufficient. Similarly, only detailed grading and nodal status should be maintained in the table.

Patients who underwent surgery and post-operative radiation therapy must be excluded.

Similar to figure 3, survival curves in HPV+ patients should be added.

Since a lot of parameters were analyzed, a multivariate analysis must be performed. Moreover, concomitant chemotherapy must be added as a confounding factor and used in the analyses.

Radiation doses should be reported.

When was HPV16 DNA analysis in plasma performed and why? Furthermore, the authors should explain how was it used in the study.

Reviewer 3 Report

The authors retrospectively analyzed 127 patients with HPV(-) and HPV(+) oropharyngeal cancer (OPC) treated using RT/CRT. Statistical analyses demonstrated that HPV(+) OPC patients had a tendency to lose their nutritional condition through RT/CRT. They conclude that HPV(+) OPC  patients need more intensive nutritional support.

Their report involves some logical and statistical problems as following.

Their conclusion should be an inevitable result, because their HPV(+) OPC patients were treated more dominantly with CRT than HPV(-) OPC patients (Table 1).

Table 2-6 have multiple comparisons problem. Multiple testing correction must be performed, e.g. using Bonferroni correction. (Table 1 is not needed to be corrected, because they are patients' background.)

Table 4, logically, p value can not be zero. Expression should be corrected.

Multivariate analyses must be performed to certify and conclude that some nutritional factors are related to OS or DFS in HPV(-)/(+) OPC patients. 

Collectively, their results can not support their conclusions enough because of multiple comparisons problem in repeated univariate statistical analyses.

As for details,

Line 45, DSF should be DFS.

In 2.3. Study design, line 136-137, the explanation for DFS is wrong. It is DSS.

Line 133, disease-specific survival (DFS) should be disease-free survival.

In tables, many digits are dislocated. They should be arranged.

In Table 2, HPV(+) N0-1, RT plus CRT can not equal 100 (%).

Line 200, LOS. In line 202, LOW. They should be unified.

In Table 4, BMI 0 is repeated. It should be BMI 1.

Similarly, In Table 5, Haemoglobin 0 is repeated.

Line 375, What is CWL?